# Accounting for Aerosols Effect in GHGSat Methane Retrieval

Qiurun Yu[1], Dylan Jervis[2], Yi Huang[1]

[1]Department of Atmospheric and Oceanic Sciences, McGill University, Montréal, QC, H3A 0B9, Canada
[2]GHGSat, Inc., Montréal, QC, H2W 1Y5, Canada

*Correspondence to*: Qiurun Yu (qiurun.yu@mail.mcgill.ca)

**Abstract.** GHGSat comprises a constellation of high spatial and spectral resolution satellites, specializing in monitoring methane emissions at 1.65 μm. This study investigates the ability to accurately retrieve both the methane mixing ratio enhancement ($\Delta X_{CH_4}$) and aerosol optical depth (AOD) simultaneously from simulated GHGSat observations that incorporate angle-dependent scattering information. Results indicate that the sign of $\Delta X_{CH_4}$ bias when neglecting aerosols changes from

negative to positive as surface albedo increases, which is consistent with previous studies. Bias in $\Delta X_{CH_4}$ is most pronounced when AOD is not simultaneously retrieved, ranging from -3.0% to 6.3% with a 0.1 AOD, a 60° solar zenith angle, and a 0.2 surface albedo for the nadir-only retrieval. Using multiple satellite viewing angles during the GHGSat observation sequence with a scattering angle ranging from 100° to 140°, the study shows that the mean bias and standard deviation of $\Delta X_{CH_4}$ are within 0.3% and 2.8% relative to the background. The correlation between simultaneously retrieved $\Delta X_{CH_4}$ and AOD shifts

from positive to negative as surface albedo increases and aerosol asymmetry factor decreases, signifying a transition of the dominating aerosol effect from aerosol-only scattering to aerosol-surface multiple scattering. The variety of scattering angle ranges has little impact on the performance of the multi-angle viewing method. This study improves the understanding of the aerosol impact on the GHGSat $\Delta X_{CH_4}$ retrieval and provides guidance for improving future GHGSat-like point-source imagers.

## 1 Introduction

Aerosols can modify photon path length via their scattering and absorption effects and have been identified as one of the major sources of errors when retrieving greenhouse gases from spectrally resolved backscattered solar radiation in the shortwave infrared (SWIR) (Aben et al., 2007; Butz et al., 2009; Connor et al., 2016; Chen et al., 2017; Huang et al., 2021). Accurately assessing greenhouse gas emissions in the presence of aerosols remains a challenge. This is because unaccounted aerosols can either enhance or reduce the absorption of light by gases, depending on factors such as aerosol concentration, aerosol height

distribution, viewing geometry, and surface albedo, among others (Butz et al., 2009; Frankenberg et al., 2012; Sanghavi et al., 2020). Houweling et al., (2005) analyzed Scanning Imaging Absorption Spectrometer for Atmospheric Chartography (SCIAMACHY) measurements of total column $CO_2$ over the Sahara and found that the unrealistically large $CO_2$ variability of 10% (37 ppm) of the total column was caused by mineral dust aerosols. Butz et al. (2009) found that if aerosols were not considered, atmospheric $CO_2$ retrieval errors larger than 1% may occur when using the SCIAMACHY and Greenhouse gases

Observing SATellite (GOSAT)-like observers. These errors are dependent on both surface albedo and the type of aerosols present. Huang et al. (2020) simulated Airborne Visible/Infrared Imaging Spectrometer – Next Generation (AVIRIS-NG) measurements for methane emissions. Their results show an underestimation of $CH_4$ resulting from aerosols, particularly those with high single scattering albedo and low asymmetry factor (such as water-soluble aerosols). These studies, among many others, underlined the importance of understanding the effect of aerosols on the remote sensing of greenhouse gases.

To account for the atmospheric scattering in SWIR satellite retrieval of greenhouse gas, a 'full-physics' retrieval requires simultaneously solving for the vertical profile of gas concentration, aerosol extinction, and the surface reflectivity by inversion of the radiance spectrum using a radiative transfer model (Butz et al., 2012; Jacob et al., 2022). However, this method is time-consuming and is likely to fail if the atmosphere is heavily polluted or if the surface is too dark (Lorente et al., 2021). In contrast to 'physics-based' methods, some proxy methods, which are much faster than full-physics retrieval and achieve similar

precision and accuracy, have been proposed. To simultaneously retrieve the $CO_2$ total column and aerosol properties, the '3-band' retrieval exploits measurements of the absorption bands of $O_2$ (0.77 μm) and $CO_2$ (1.61 μm and 2.06 μm) to retrieve aerosol amount, height distribution, and size distribution based on a simple aerosol microphysical model (Butz et al., 2009). However, this approach requires additional consideration of the uncertainty of a prior estimate of $CO_2$ (Butz et al., 2012). According to Parker et al. (2020), methane mixing ratio ($X_{CH_4}$) can be retrieved by using both $CH_4$ (1.65 μm) and the adjacent

$CO_2$ band (1.61 μm) by taking advantage of the $X_{CH_4}/X_{co_2}$ ratio without accounting for atmospheric scattering. However, this '$CO_2$ proxy' method is subject to bias for sources that co-emit $CH_4$ and $CO_2$ such as gas flaring. Depending on the instrument design and its limitations, the approach to accounting for the effect of aerosols on greenhouse gas retrieval varies.

       GHGSat, Inc. has developed a nano-satellite system that measures greenhouse gas emissions from individual industrial facilities (Varon et al., 2019). Its satellite achieves a combination of fine spatial resolution and spectral resolution by pointing

at targeted methane point sources (Jervis et al., 2021; Jacob et al., 2022). As of the time of writing, GHGSat has launched a constellation of 11 commercial satellites (GHGSat-C1 to C11), which monitors methane emissions from natural gas industry operations, landfills, hydroelectric reservoirs, and oil sands operations among others (Calvello et al., 2017; Varon et al., 2019; Jacob et al., 2022; Maasakkers et al., 2022). However, industrial activities such as oil extraction and pre-treatment involve not only gaseous emission but also aerosol production (e.g., water-soluble and black carbon aerosols). The continued development

of the GHGSat satellite requires identifying and minimizing the uncertainty in methane retrieval due to aerosol interference. Their newer satellites only target the $CH_4$ band; consequently, the above-mentioned 'proxy' methods to account for the aerosol effects do not apply to their instrument. An accurate GHGSat aerosol retrieval model for GHGSat would not only reduce the

uncertainty in their methane retrieval but also provide a new, aerosol data product, potentially making a high spatial resolution air quality measurement from the space.

The angular dependence of aerosol scattering allows space-borne observations of aerosol properties based on multi-angle measurements, providing the potential to mitigate aerosol-induced errors in current greenhouse gas satellite observations. Frankenberg et al. (2012) demonstrated that adding multiple satellite viewing angles to the Orbiting Carbon Observatory 2 (OCO-2)-like observations enhances the ability to retrieve aerosol properties. The aerosol information can in turn significantly decrease errors in the measurement of $CO_2$ and $CH_4$ total columns. However, this multi-angle viewing method was applied to

area flux mappers which are designed to observe emissions on regional scales. There has been little study demonstrating how to retrieve aerosols using point source imagers like GHGSat. A method to co-retrieve aerosols and methane using GHGSat spectral content could address a gap in current research on point source imagers, improve the accuracy of their greenhouse gas retrieval, and provide greater details about aerosol and methane concentrations locally.

    This study has three objectives. First, we assess how aerosols impact the accuracy of GHGSat methane mixing ratio

enhancement ($\Delta X_{CH_4}$) retrieval when the aerosols are present but not retrieved. This assessment involves simulating GHGSat satellite observations for a wide range of aerosol optical properties and surface albedo values to evaluate the distribution and magnitude of any resulting bias in $\Delta X_{CH_4}$ under different aerosol and surface conditions. Second, we simultaneously retrieve aerosol optical depth (AOD) and $\Delta X_{CH_4}$ using a multi-angle viewing method in comparison with the $\Delta X_{CH_4}$-only retrieval under the same conditions. Finally, we investigate how different scattering angles as well as uncertainties in aerosol type, height

distributions, and surface albedo affect the performance of the simultaneous retrieval.

    This paper is organized into five sections. Section 2 provides an overview of the atmospheric models, GHGSat instrument model, and the simultaneous retrieval methods for aerosols and methane. Section 3 evaluates the errors in GHGSat methane retrieval under various aerosol, surface, and satellite zenith angle conditions. Synthetic data is used to conduct retrieval under two scenarios: methane-only nadir retrieval and the simultaneous retrieval of methane and aerosols using the multi-angle

viewing method. Section 4 investigates the impact of satellite viewing angles as well as the uncertainty in aerosol and surface albedo on simultaneous retrieval. A summary is presented in Section 5.

## 2 Method

### 2.1 Atmospheric Model

    The Top of the Atmosphere (TOA) radiance detected by the satellite comes from both the direct and diffuse reflections. The

incoming sunlight is reflected to space by the Earth's surface and atmospheric scatterers such as aerosols. When the solar beam travels through the atmosphere, it can partly be absorbed along its path by atmospheric absorbers, such as methane molecules

and aerosols. Additionally, multiple scattering processes occur between the surface and aerosol layers. To assess the radiative impact of aerosols in the GHGSat methane retrieval, a forward model is required to simulate GHGSat-measured solar radiation. The radiative transfer forward model of this study is DIScreet Ordinate Radiative Transfer (DISORT) version 4.0.99 (Stamnes et al., 1988). As one of the most general and versatile plane-parallel radiative transfer models, DISORT has been widely used for the remote sensing of greenhouse gases, aerosols, and clouds (Tzanis and Varotsos 2008; Wang et al. 2013; Boiyo et al. 2019). It can numerically compute satellite-measured radiance at different wavenumbers using discrete vertical coordinates. For each atmospheric layer, the spectral optical depth and single scattering albedo for atmospheric molecules are computed by using a rigorous line-by-line radiative transfer model (LBLRTM) over a 0.1 $cm^2$ interval (Clough et al., 2005). The mid-latitude summer profile is chosen as the default atmospheric state. The absorption of four main atmospheric absorptive gases ($H_2O$, $CO_2$, $O_3$, $CH_4$) at 45 layers is considered through line-by-line calculations.

To facilitate the analysis of aerosol-induced errors during the GHGSat $CH_4$ retrieval, this study focuses on the shortwave near-infrared band (1662 - 1672 nm). These bands cover the absorption lines which are mainly caused by $CH_4$. The surface is assumed to be Lambertian and we adopt the 16-stream approximation. With the specified viewing geometry and surface albedo, DISORT can calculate the solar radiation backscattered to space by the Earth's surface and atmosphere. For a clean atmosphere with a surface albedo of 0.2, the TOA upward radiance simulated by DISORT is shown in Fig. 1b. The solar zenith angle is 60°, and the satellite field of view is in the nadir position. In Fig. 1b, strong $CH_4$ absorptions are observed around 1666 nm, consistent with results from other studies like Jervis et al. (2021) and Chan Miller et al. (2023). Given that GHGSat measures methane concentrations by analyzing spectrally decomposed solar backscattered radiation within the methane absorption band ($\sim 1.65\mu m$), this alignment supports the adequacy of DISORT-simulated radiance for capturing the methane effect. With the TOA incoming solar radiance known (Fig. 1a), the TOA reflectance ($Ref_\lambda^{TOA}$) can be calculated following:

$$Ref_\lambda^{TOA} = \frac{radiance_\lambda^{TOA\uparrow}}{radiance_\lambda^{TOA\downarrow}} , (1)$$

where $radiance_\lambda^{TOA\downarrow}$ and $radiance_\lambda^{TOA\uparrow}$ are the TOA downward and upward radiance at the wavelength $\lambda$. The radiance is in unit $Wm^{-2}sr^{-1}m^{-1}$. For GHGsat retrieval only considering gas absorbers, the relative depth of the absorption line directly corresponds to the retrieved methane enhancement compared to the background. Therefore, $Ref_\lambda^{TOA}$ is directly linked to the retrieved $CH_4$ enhancement and is shown in Fig. 1c.

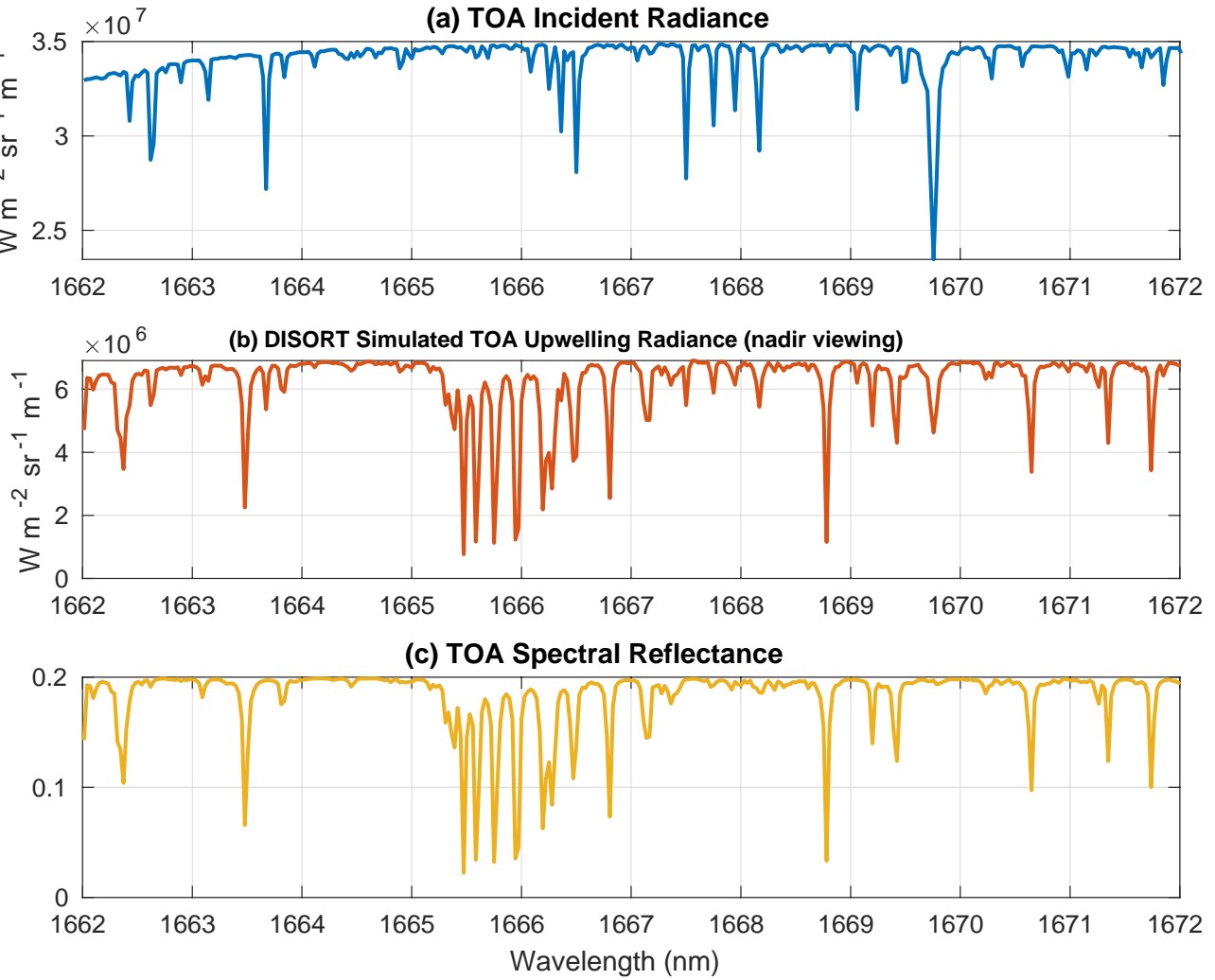

**Figure 1. (a) TOA incoming solar radiance; (b) Simulated TOA upward radiance (nadir viewing); (c) Spectral reflectance (nadir viewing). Spectra are simulated with a surface albedo of 0.2 and a solar zenith angle of 60°.**

### 2.2 Aerosol Settings

Many factors such as aerosol type, concentration, and height distribution can impact the radiance measurement. In this study, the aerosol types are predefined in the retrieval. We used climatological aerosol optical property values from Ayash et al. (2008) to account for the diverse range of particles found in industrial sites. For aerosols composed of multiple components, the single scattering albedo (SSA) spans from 0.86 to 0.98, while the asymmetry factor (g) ranges from 0.54 to 0.76. GHGsat mainly focuses on measuring $CH_4$ enhancement over methane hotspots, where $CH_4$ and the co-emitted aerosols are concentrated near the surface. To emulate the aerosol emissions from the industrial plume, one arbitrary aerosol layer is added near the surface between 1000 to 900 hPa. Considering the instrument limitation of one spectral band, the simplified treatment of aerosols in the forward model allows for a more direct physical interpretation of the effect of aerosols on methane retrieval.

We focus mainly on the AOD retrieval because this variable is highly representative of the aerosol radiation effect (Frankenberg et al., 2012; Yu and Huang, 2023a, b). In this study, the simulated truth of AOD is 0.1 at SWIR (~0.3 AOD at 550nm). This threshold is selected in the retrieval because it is used as filter values in other $XCH_4$ retrieval studies (Lorente et al., 2021).

## 2.3 The Multi-angle Viewing Method

The multi-angle aerosol retrieval method proposed by Frankenberg et al. (2012) uses the radiance difference at various viewing geometries to retrieve aerosol information and takes advantage of the fact that aerosols scatter more light forward than backward. In this study, satellite azimuth angles are chosen as 0° and 180° to represent the forward-viewing and backward-viewing observations (i.e., straight south and north-looking), respectively. Table 1 summarizes the angles used in the multi-angle viewing simulations. The scattering angle $\Theta$ is calculated following (Thompson et al., 2022) :

$$\Theta = 180° - \arccos[\cos\theta_1\cos\theta_2 + \sin\theta_1\sin\theta_2\cos(\varphi_1 - \varphi_2)] \quad (2),$$

where $\theta_1$ and $\theta_2$ are solar and satellite zenith angles, $\varphi_1$ and $\varphi_2$ are solar and satellite azimuth angles, respectively. Fig. 2 shows the schematics of the multi-angle viewing method and its corresponding angles. This study assumes the Henyey-Greenstein Phase Function for aerosols (Toublanc, 1996), which defines the phase function following:

$$P_{HG}(\cos\Theta) = \frac{1-g^2}{(1-2g\cos\Theta+g^2)^{3/2}} \quad (3),$$

where g is the aerosol asymmetry factor. The high g value implies that most of the scattered light is directed forward in the same general direction as the incident light.

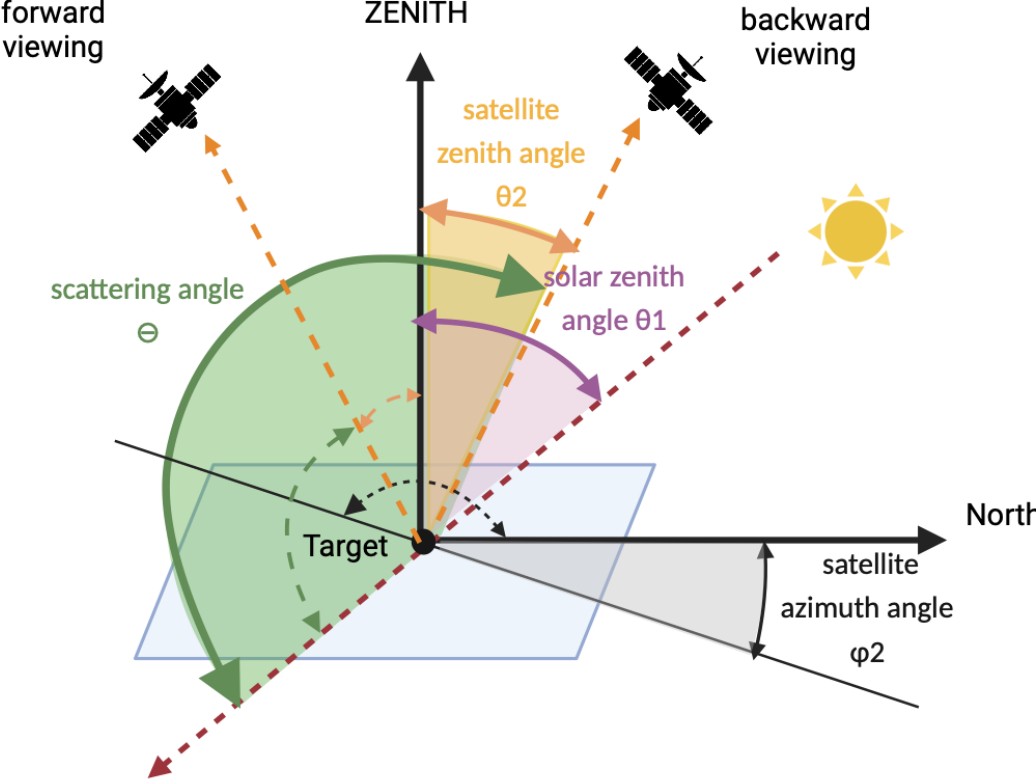

**Figure 2. Schematic of a given solar and viewing geometry, as well as corresponding scattering angle for forward and backward viewing modes. Solar zenith angle $\theta_1$, satellite zenith angle $\theta_2$, and satellite azimuth angles $\varphi_2$ are indicated by the purple, orange, and black double arrow curves. Scattering angle Θ is represented by the green double arrow curves. The viewing angles are depicted using solid and dashed double-arrow curves for the backward and forward viewing modes, respectively. In this case, the satellite azimuth angles are 0° and 180° for the backward and forward viewing directions (angles relative to the north-facing vector).**

**Table 1 Angles used in the multi-angle satellite viewing simulations for Sect. 3.1 and 3.2**

|  | Solar zenith angle $\theta_1$ | Satellite zenith angle $\theta_2$ | Solar azimuth angle $\varphi_1$ | Satellite azimuth angle $\varphi_2$ | Scattering angle Θ |
|---|---|---|---|---|---|
| Forward viewing | 60° | 20° | 180° | 0° | 100° |
| Nadir | 60° | 0° | 180° | 0° | 120° |
| Backward viewing | 60° | 20° | 180° | 180° | 140° |

## 2.4 GHGSat Instrument Model

A nominal GHGSat measurement covers a targeted 12×15 km$^2$ area with approximately 25×25 m$^2$ pixel resolution and 0.3 nm spectral resolution (Jervis et al., 2021; Jacob et al., 2022). The instrument adjusts its altitude to ensure that the targeted area remains within its field of view for an extended period, thereby enhancing its signal-to-noise ratio (SNR). During the
observation sequences, the GHGSat spectrometer typically takes 200 images of closely overlapping atmospheric absorption spectrum. A more detailed description of the GHGSat instrument design and its measurement concept is presented in Jervis et al. (2021). To simulate GHGSat measurements, this study focuses on the spectral region between 1662 nm and 1672 nm and applies a Gaussian broadening kernel of 0.3 nm full width at half maximum (FWHM). Using the multi-angle viewing method, the satellite observes the target position from different angles, transitioning from a forward view to a view looking directly
downward (nadir), and finally to a backward view.

As an example, Fig. 3 displays the simulated GHGSat radiance corresponding to the solar geometry detailed in Table 1, under the assumption of a single layer of sulfate aerosols near the surface with an SSA of 1 and a g of 0.78. These simulations are based on a surface albedo of 0.2 and an AOD of 0.1 at SWIR for illustration purposes. Fig. 3 indicates that with the addition of a highly reflective aerosol layer, TOA reflectance in the forward viewing direction exceeds that in the nadir or backward
viewing direction. This suggests the importance of viewing angles in GHGSat observations when aerosols are present and highlights the potential for retrieving them using multi-angle information. In the following discussions, a positive satellite zenith angle corresponds to an azimuth angle of 0° (forward viewing), while a negative zenith angle corresponds to an azimuth angle of 180° (backward viewing).

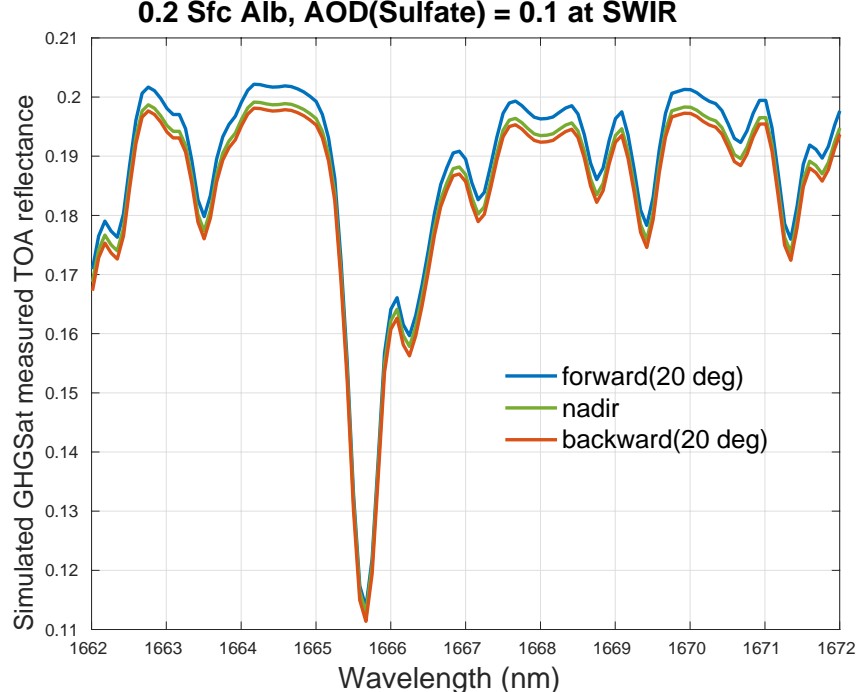

**0.2 Sfc Alb, AOD(Sulfate) = 0.1 at SWIR**

**Figure 3. Simulated TOA reflectance measured by GHGSat instrument at a spectral resolution of 0.3 nm FWHM. The instrument observes the surface with an albedo of 0.2 from different viewing positions: forward viewing, nadir, and backward viewing following Table 1. Sulfate aerosols with 0.1 AOD at SWIR are added near the surface.**

**2.5 Retrieval Methods**

Fig. 4 illustrates the steps of the simulated retrieval process in this study. First, we combine the atmospheric molecule optical

properties calculated from LBLRTM with the aerosol optical properties to run the atmospheric model (DISORT). Then DISORT is further modified according to the GHGSat instrument design to build a complete forward model $F(X)$ to simulate the TOA reflectance (Eq.(1)). $X$ is the state vector, which includes elements such as methane mixing ratio $X_{CH_4}$, aerosol optical depth AOD, and the surface albedo $X_{alb}$. The goal of the retrieval is to estimate $\Delta X_{CH_4}$ and $X_{alb}$ for the $\Delta X_{CH_4}$-only retrievals, and to estimate $\Delta X_{CH_4}$, AOD, and $X_{alb}$ for the simultaneous retrieval using the multi-angle viewing method from the

measurement vector $y$:

$$y = F(X) + \epsilon_y \ (4)$$

where $\epsilon_y$ is the measurement error.

Full GHGSat retrieval consists of two steps: a scene-wide retrieval to estimate the background average state vector $\widehat{X}$ and a per-cell retrieval to estimate the local methane plume enhancement. Note that surface albedo is retrieved in both cases. In this

study, we focus on the per-cell retrieval assuming known background $\widehat{X}$. In Jervis et al. (2021), a linearized forward model (LFM) is proposed for the GHGSat spatially resolved $\Delta X_{CH_4}$-only retrieval.

$$\boldsymbol{F}^{LFM}(\boldsymbol{X}) = (X_{alb} + b_1 n + b_2 n^2)\left[\boldsymbol{F}(\widehat{\boldsymbol{X}}) + (X_{CH_4} - \widehat{X_{CH_4}})\widehat{K_{X_{CH_4}}}\right]$$

$$= (X_{alb} + b_1 n + b_2 n^2)\left[\boldsymbol{F}(\widehat{\boldsymbol{X}}) + \Delta X_{CH_4}\widehat{K_{X_{CH_4}}}\right] \quad (5)$$

$\widehat{X}$ is the linearization point, at which the state vector in the observation scene is assumed to be in the background state. $\boldsymbol{K}_{\widehat{X}}$, the
Jacobian that corresponds to different state vector elements, is a matrix of partial derivatives that describes how the simulated TOA reflectance changes with respect to the elements of the state vector.

$$\boldsymbol{K} = \frac{\partial \boldsymbol{F}(\boldsymbol{X})}{\partial \boldsymbol{X}} \quad (6)$$

To account for the bidirectional distribution of surface albedo and the per-pixel signal changes resulting from satellite motion, the forward model includes a second-order polynomial that is a function of the image frame index n (Jervis et al., 2021). In
this study, we employed the LFM model as current GHGSat instruments and estimated $\Delta X_{CH_4}$ and $X_{alb}$ by minimizing the difference between the simulated instrument-measured $\boldsymbol{y}$ and $\boldsymbol{F}^{LFM}(\boldsymbol{X})$.

For simultaneous $\Delta X_{CH_4}$ and AOD retrieval, we added AOD as an additional variable of interest in the LFM as depicted below.

$$\boldsymbol{F}^{LFM} = (X_{alb} + b_1 n + b_2 n^2)\left[\boldsymbol{F}(\widehat{\boldsymbol{X}}) + \Delta X_{CH_4}\widehat{K_{X_{CH_4}}} + AOD\widehat{K_{AOD}}\right] \quad (7)$$

The applicability of the simultaneous $\Delta X_{CH_4}$ and AOD retrieval method mainly comes from two aspects: enhancing the
methane gas retrieval accuracy by accounting for aerosols effect for GHGSat-like point source imagers and measuring aerosol plumes using such imagers. By integrating LBLRTM, DISORT, and GHGSat instrument model and applying the same inverse model (Eq. 5) utilized in current GHGSat operations, our retrieval results can provide a truthful assessment of the simultaneous $\Delta X_{CH_4}$ and AOD retrieval technique on GHGSat-like point source imagers using the multi-angle viewing method. In the following section, the retrieval method is tested across a wide range of aerosol optical properties, surface albedo, and satellite
zenith angle conditions, demonstrating its direct applicability to real measurements.

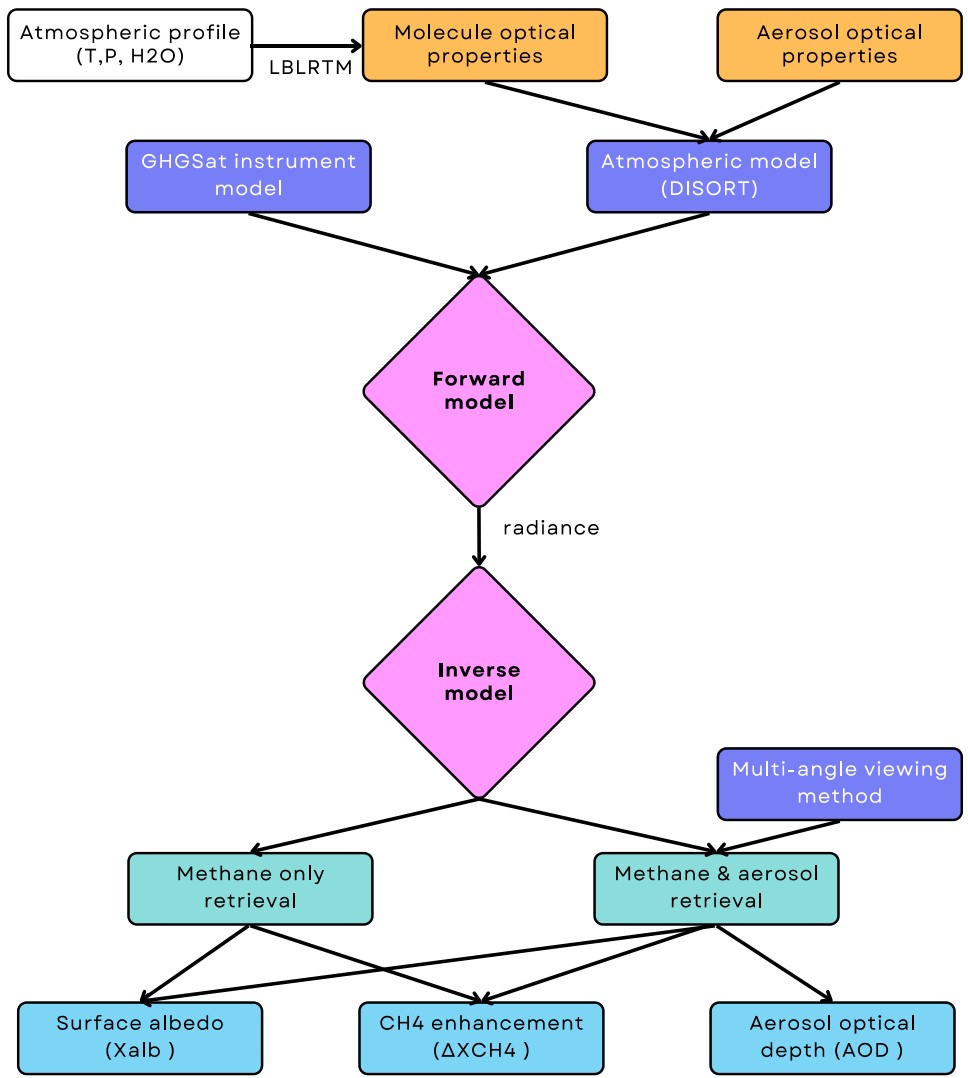

**Figure 4. Schematic diagram of retrieval steps.**

## 3. Assessment of Two Retrieval Methods

This paper aims to estimate the impact of aerosols on GHGSat methane retrieval, assess the validity of the multi-angle viewing method for the GHGSat aerosol and methane co-retrieval, and understand the algorithm's sensitivity to different input parameters, including surface albedo, SSA, g and satellite geometry. To achieve this, retrieval experiments were conducted using synthetic data, and the retrieval errors were estimated.

Fig. 5 depicts Jacobians with respect to the methane mixing ratio and AOD with different SSA and g values when the surface albedo is 0.2 and the solar zenith angle is 60°. A negative $K_X$ value indicates that the reflectance at the TOA decreases as the value of the state vector element X increases. As expected, $K_{CH_4}$ is negative considering the absorption properties of methane. Similarly, $K_{AOD}$ is also negative in the case of absorbing aerosols (SSA=0.1). For strongly scattering aerosols (SSA=0.95) with high g (0.7) over the dark surface (0.2), $K_{AOD}$ is slightly positive at the forward viewing position and negative at the backward viewing position (Fig. 5b). When the satellite is at the backward viewing position, the aerosol-only scattering is less pronounced because less light scatters towards the space in that direction, resulting in a negative $K_{AOD}$. In contrast, in the forward viewing position, more light is scattered by aerosols toward the space, and this effect prevails over the effect of atmospheric absorption enhancement due to aerosol-surface multiple scattering, resulting in a slightly positive $K_{AOD}$. This is particularly noticeable when the asymmetry factor, g, is low (0.1). In this case, the dominant factor is the shortening of the light path caused by aerosol-only scattering, which leads to a positive $K_{AOD}$ regardless of the viewing angle (Fig. 5c). For aerosol with low g (0.1) over mid-range albedo (0.5), the competition between the aerosol-only scattering and aerosol-surface multiple scattering result in a near zero $K_{AOD}$ (Fig. 5e).

Fig. 5 also compares the Jacobians between satellite forward (scattering angle 100°) and backward (scattering angle 140°) viewing positions. With high SSA and g values, differences in aerosols Jacobian between the two angles increase, providing more information to the simultaneous retrieval. For simulated GHGsat retrieval using the multi-angle viewing technique, the scattering angle increases from 100° to 140° from forward viewing to backward viewing as depicted in Fig. 6a. Given a specific asymmetry factor value (g=0.78), the angular distribution of aerosol scattering energy within this scattering angle range is depicted in Fig. 6b. It illustrates that the intensity of scattering energy diminishes as the scattering angle increases, leading a decrease in TOA reflectance. The greater the variation in TOA reflectance at various angles, the richer the aerosol information it can provide for simultaneous retrieval.

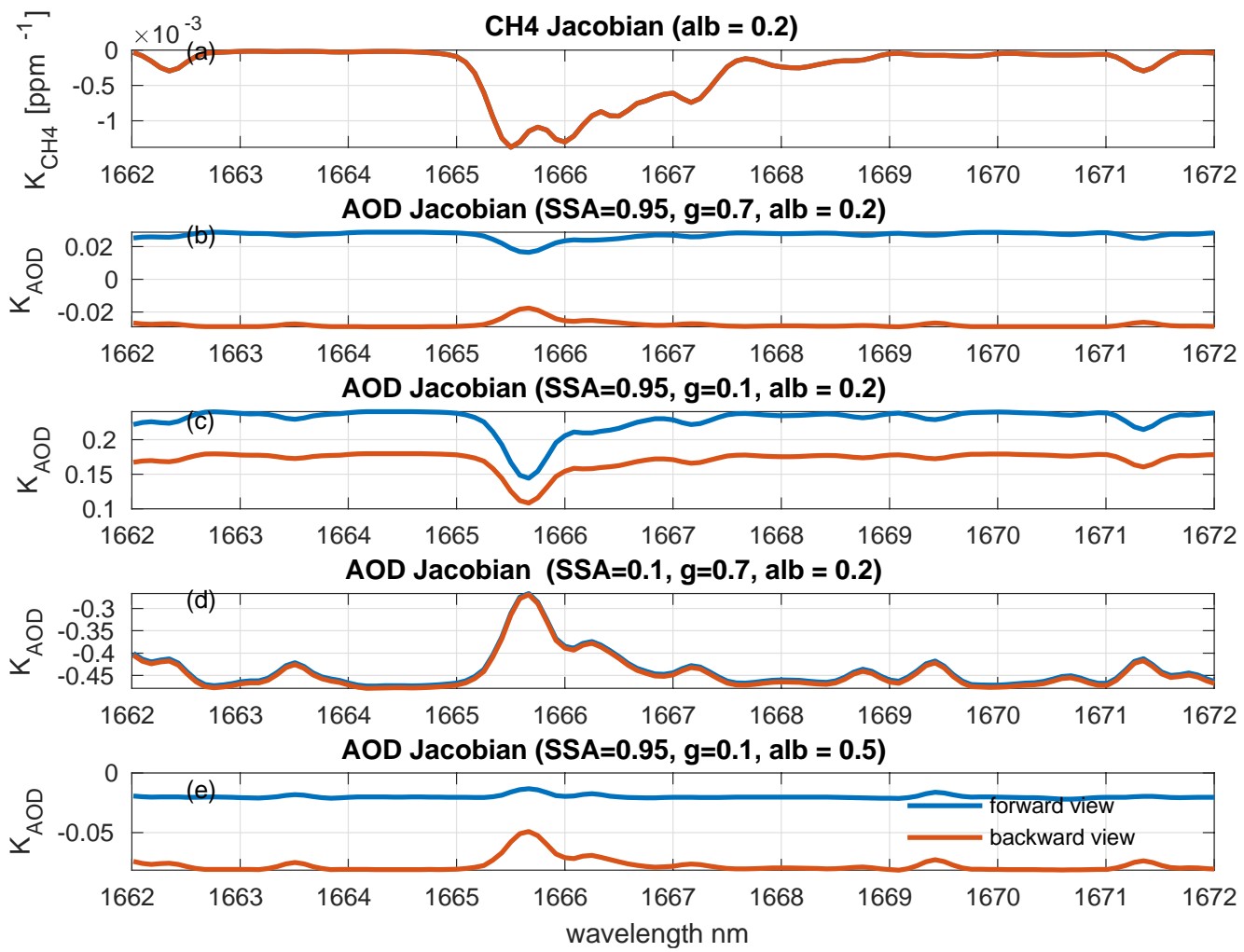

Figure 5. Jacobian of TOA reflectance with respect to (a) methane mixing ratio; (b) AOD with an SSA of 0.95, a g of 0.7, and a surface albedo of 0.2; (c) AOD with an SSA of 0.95, a g of 0.1, and a surface albedo of 0.2; (d) AOD with an SSA of 0.1, a g of 0.7, and a surface albedo of 0.2; (e) AOD with a SSA of 0.95, a g of 0.1, and a surface albedo of 0.5. Aerosols are concentrated near the surface and the forward and backward viewing angle settings follow Table 1.

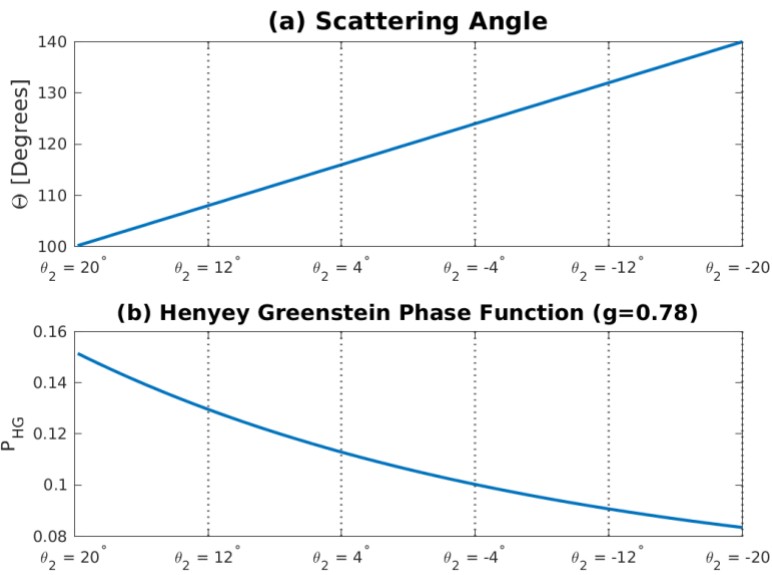

**Figure 6. (a) Scattering angles Θ and (b) Phase function $P_{HG}$ for g =0.78 as a function of the satellite zenith angle $\theta_2$ during GHGSat observation sequence when applying multi-angle viewing method with a maximum satellite zenith angle of 20°.**

As instrument measurements are always subject to noise and errors, it is important to include these in the simulated retrieval process to represent real-world conditions. During the simulated retrieval, white noise and 1/f errors with a magnitude of 0.2% each (calculated as the standard deviation of the individual noise fields) are added to the TOA reflectance. The background value for the methane mixing ratio is 1.7 ppm. The simulated truth of methane enhancement ($\Delta X_{CH_4}$) and aerosol optical depth (AOD) are 0.1 ppm and 0.1, respectively. We performed 1000 independent retrieval for each aerosol and surface albedo setting and we quantified the mean bias and standard deviation of retrieved $\Delta X_{CH_4}$ relative to the background to represent the level of accuracy and consistency of retrieved data.

**3.1 The Impact of Incorporating AOD and Employing the Multi-angle Viewing Method**

To assess the extent to which incorporating aerosols and applying the multi-angle viewing method can improve the GHGsat methane retrieval, we conducted retrieval under four conditions: when aerosols are present but not retrieved for the (1) nadir-only methane retrieval and (2) the multi-angle viewing methane retrieval, and when aerosols and methane are co-retrieved (3) in the nadir viewing mode and (4) in the multi-angle viewing mode. Mean bias in the retrieved $\Delta X_{CH_4}$ and AOD are shown in Fig.7.

Fig. 7a and 7b indicate that the multi-angle viewing method alone has little impact on the methane retrieval accuracy for the methane-only retrieval. For extreme aerosol SSA and g values, the mean bias in $\Delta X_{CH_4}$ ranges from 6% to -25% when aerosols

are neglected in the retrieval. After adding AOD as an additional retrieval variable, the mean bias in $\Delta X_{CH_4}$ significantly decreased to 0.32% (Fig. 7c). Further applying the multi-angle viewing method with angles specific in Table 1 reduced the mean bias in $\Delta X_{CH_4}$ even further to 0.15% (Fig. 7d). This suggests that the good performance of aerosol and methane co-retrieval using the multi-angle method largely comes from incorporating AOD as an additional retrieval variable.

As for the AOD retrieval performance, Fig. 7e and 7f suggest that applying the multi-angle viewing method yields better accuracy in the AOD retrieval than the nadir-only method, with the mean bias in AOD being less than 0.02. In theory, the multi-angle viewing method should provide more information than nadir viewing observations, especially for aerosol retrieval. The relatively modest improvement observed with the multi-angle viewing method in our study compared to the substantial enhancement achieved by adding AOD alone may stem from the instrumental limitation of intensity-only measurements within a single spectral band. Nevertheless, our study continues to employ the multi-angle viewing method for simultaneous aerosol and methane retrieval, as it yields the most significant improvement in retrieval accuracy and precision for both $\Delta X_{CH_4}$ and AOD.

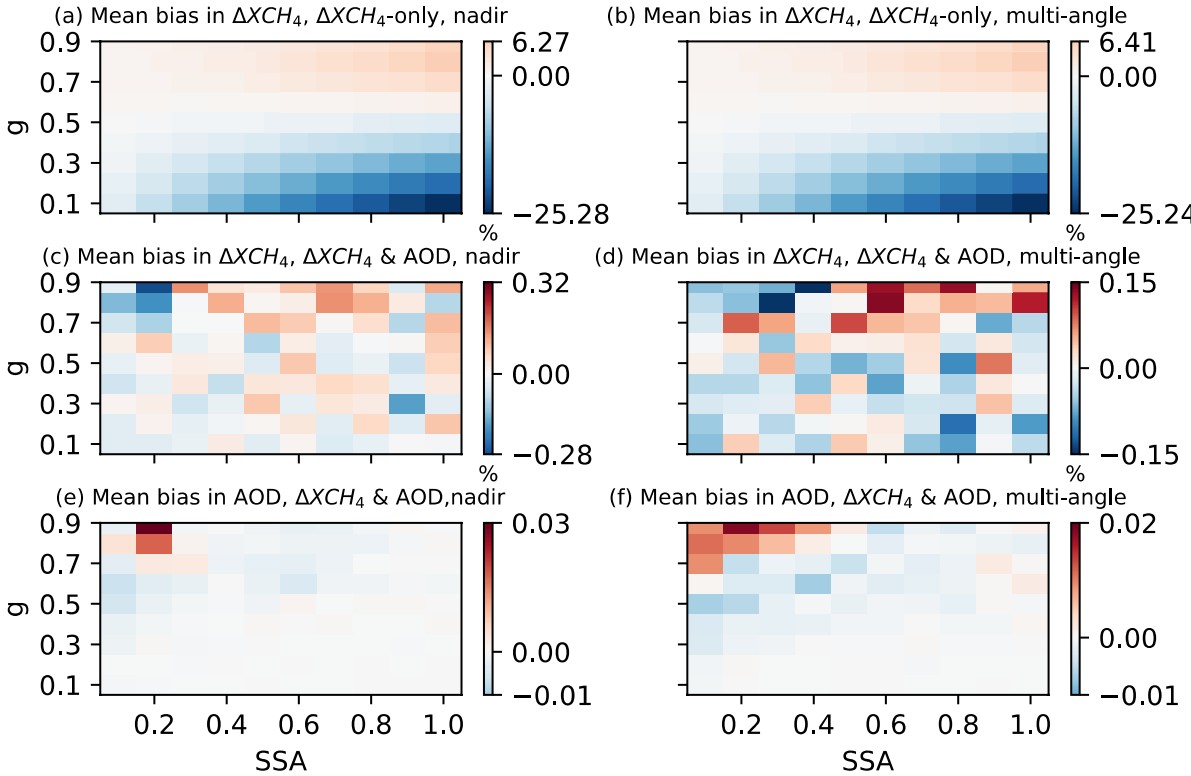

**Figure 7.** Left column: nadir-only viewing mode; Right column: multi-angle viewing mode (Table 1, scattering angle ranges from 100°-140°); Upper row: mean bias in retrieved $\Delta X_{CH_4}$ values when aerosols are present but not retrieved; Mid row: mean bias in retrieved $\Delta X_{CH_4}$ values when aerosols and methane are simultaneously retrieved; Lower row: mean bias in retrieved AOD values when aerosols and methane are simultaneously retrieved; Retrieval results are

displayed as a function of aerosol SSA and g when surface albedo is 0.2. The simulated truth of $\Delta X_{CH_4}$ and AOD are 0.1 ppm and 0.1, respectively. The mean bias in $\Delta X_{CH_4}$ is calculated relative to the background methane mixing ratio.

### 3.2 Comparisons between the $\Delta X_{CH_4}$-only Retrieval and Simultaneous $\Delta X_{CH_4}$ and AOD Retrieval

To examine the performance of different retrieval methods, we conducted simulated retrieval at a range of surface albedo and aerosol optical properties. The mean bias and standard deviations of retrieved variables ($\Delta X_{CH_4}$, $AOD$, and $X_{alb}$) are compared
under two scenarios: (1) when aerosols are present but not retrieved in the nadir-viewing mode, and (2) when both $\Delta X_{CH_4}$ and AOD are retrieved simultaneously using multi-angle viewing method.

### 3.2.1 Aerosol SSA and g Impact

As we only retrieve AOD for aerosol-related parameters, unaccounted variables such as aerosol single scattering albedo (SSA) and asymmetry factor (g) can influence our results. To assess this impact, we fix the background surface albedo at 0.2 and
examine how the mean bias and STD vary with different combinations of aerosol SSA and g.

Fig. 8a and 8d display the mean bias of retrieved $\Delta X_{CH_4}$ and $X_{alb}$ values for the $\Delta X_{CH_4}$-only retrieval scenario. The angle setting follows Table 1. When retrieving $\Delta X_{CH_4}$ without accounting for aerosols, the $\Delta X_{CH_4}$-only method underestimates $\Delta X_{CH_4}$ for situations with low aerosol g and overestimates it in cases with high aerosol g. This occurs because when g is low, aerosols scatter more light back to space, reducing the absorption of $CH_4$. Conversely, when aerosol g is high, increased aerosol-surface
multiple scatterings lead to greater atmospheric $CH_4$ absorption. Fig. 8a also shows that the magnitude of retrieval bias increases with the increase of SSA. For a 0.2 surface albedo, the maximum bias in $\Delta X_{CH_4}$ for $\Delta X_{CH_4}$-only retrieval can reach - 25% relative to the background with extremely high SSA and low g values. These results are in agreement with other studies (Huang et al., 2020). Both increasing SSA and decreasing g enhance the radiation scatter back to space, thereby decreasing the atmospheric methane absorptions. For typical optical property ranges of aerosols (SSA ∈ [0.86,0.98] and g ∈ [0.54,0.76]),
mean bias in $\Delta X_{CH_4}$ falls between -3.0% to 6.3% for $\Delta X_{CH_4}$-only nadir retrieval. Neglecting aerosols also affects the retrieval of $X_{alb}$. As shown in Fig.8d, $X_{alb}$ is underestimated (overestimated) when SSA is small (large).

In contrast, Fig. 9 suggests that simultaneous retrieval of $\Delta X_{CH_4}$ and AOD can significantly improve the accuracy of $\Delta X_{CH_4}$ retrieval, while also retrieving relatively accurate values for AOD and $X_{alb}$. Using simultaneous retrieval can reduce the mean bias in $\Delta X_{CH_4}$ to within 0.1% (Table 2) for typical optical property ranges of aerosols. As for the consistency of the
simultaneous retrieval, Fig. 9d indicates that the maximum STD in $\Delta X_{CH_4}$ is near 2.5%, which is slightly higher than that in the $\Delta X_{CH_4}$-only retrieval (~1.6%). This results from the near-zero AOD Jacobian values (Fig. 5b). Although aerosols have little effect on the TOA reflectance under these conditions, their inclusion in the simultaneous retrieval inevitably increases uncertainty in retrieved $\Delta X_{CH_4}$. As for the AOD results, the mean bias falls within 1.7% for typical aerosol optical property ranges (Fig. 9b), with the STD showing a slightly high value, suggesting larger retrieval uncertainties when aerosol SSA and
g vary. In general, the multi-angle method performs better on AOD retrieval when aerosols have high SSA and high g, which

can be explained by the more pronounced AOD Jacobian differences between forward and backward viewing angles as indicated by Fig. 5b. In the retrieved surface albedo results (Fig. 9c), the mean bias in $\Delta X_{alb}$ is less than 2.1% for typical aerosol optical property ranges. The mean bias and STD distribution pattern of $X_{alb}$ are similar to those of AOD, resulting from the interference of aerosol scattering energy with surface albedo retrieval.

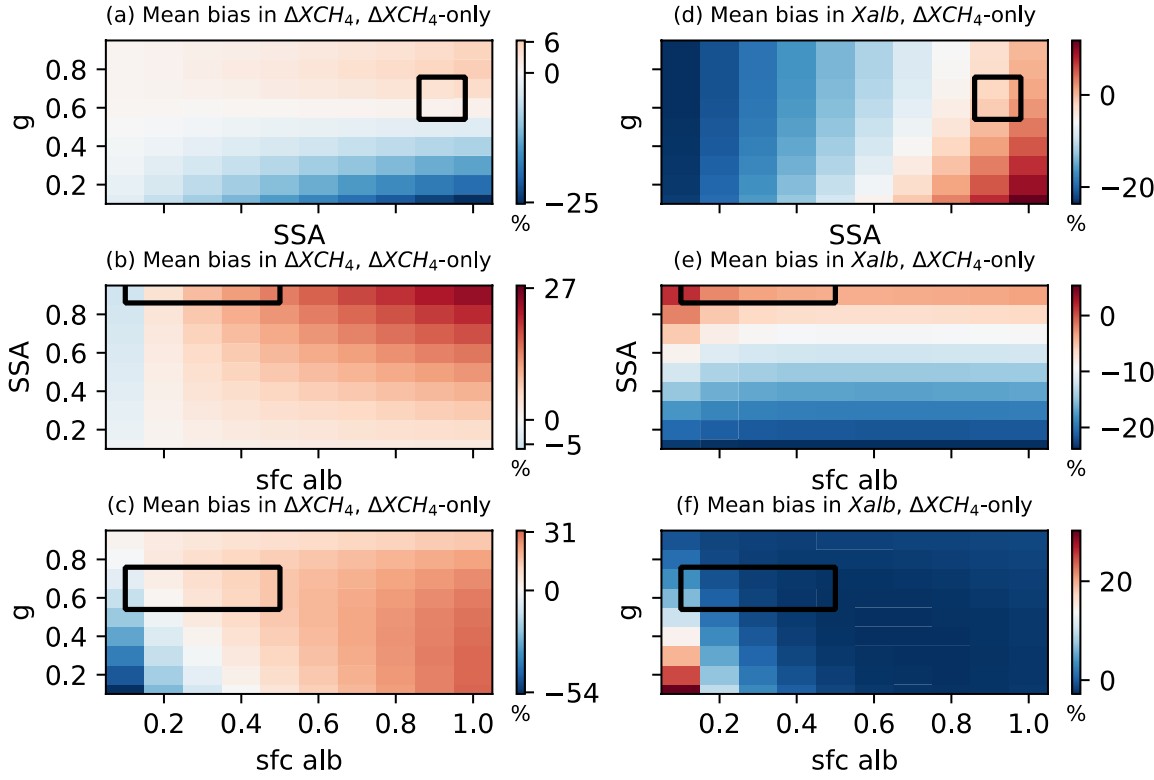


**Figure 8. Mean bias of retrieved $\Delta X_{CH_4}$ (left column) and $X_{alb}$ (right column) values when aerosols are present but not retrieved in the nadir viewing mode; Upper row: Mean bias as a function of aerosol SSA and g when surface albedo is 0.2. Middle row: Mean bias as a function of surface albedo and aerosol SSA when aerosol g is 0.7. Bottom row: Mean bias as a function of surface albedo and aerosol g when aerosol SSA is 0.95. The black box represents typical values for**

**aerosol optical property and surface albedo ranges (SSA ∈ [0.86, 0.98], g ∈ [0.54, 0.76], and sfc alb ∈ [0.1, 0.5]) in the observation. The simulated truth of $\Delta X_{CH_4}$ and AOD are 0.1 ppm and 0.1, respectively. The scattering angle ranges from 100°-140°.**

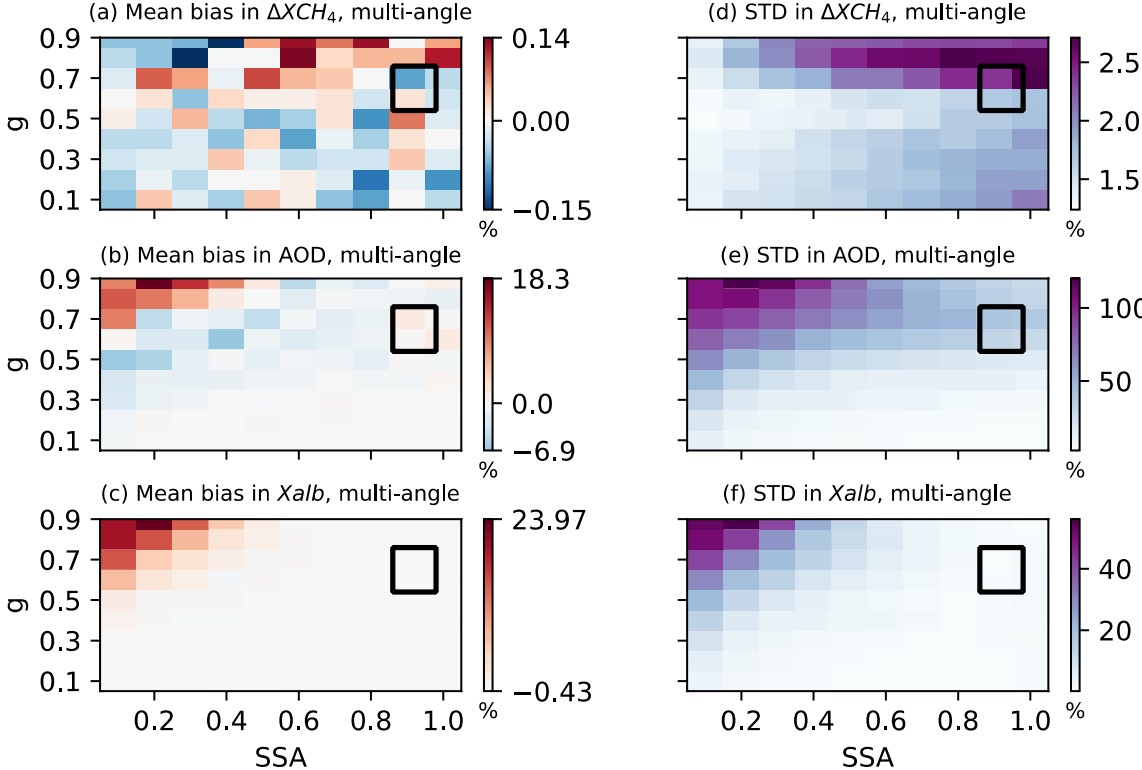

**Figure 9.** Mean bias (left column) and standard deviations (STD) (right column) of retrieved $\Delta X_{CH_4}$, AOD, and $X_{alb}$ as a function of aerosol SSA and g. The simulated truth of $\Delta X_{CH_4}$, AOD, and $X_{alb}$ are 0.1 ppm, 0.1, and 0.2, respectively. The scattering angle ranges from 100°-140°. The black box represents the typical values for aerosol optical property ranges (SSA ∈ [0.86, 0.98] and g ∈ [0.54, 0.76]) in the observation.

### 3.2.2 Surface Albedo Impact

Since the interaction between aerosols and the underlying surface can largely determine the retrieval performance, we further explored the accuracy and precision of the retrieved $\Delta X_{CH_4}$, $AOD$, and $X_{alb}$ for $\Delta X_{CH_4}$-only retrieval and simultaneous retrieval under different surface albedo conditions.

Fig. 8b and 8e display the distribution of mean bias in $\Delta X_{CH_4}$ and $X_{alb}$ for $\Delta X_{CH_4}$-only nadir retrieval when aerosol g is fixed as 0.7. As shown in Fig.8b, neglecting aerosols results in an overestimation (underestimation) of the retrieved $\Delta X_{CH_4}$ with high (low) surface albedo. These results are in agreement with other studies (Butz et al., 2009; Huang et al., 2020) despite the differences in retrieval variables, experiment settings, and instruments. High surface albedo enhances the surface and aerosol multiple scattering, leading to increased methane absorptions. Conversely, low surface albedo favors aerosol-only scattering, reducing methane absorptions. As a result, in the case of $\Delta X_{CH_4}$-only retrieval, the bias is most pronounced (~27%) when both aerosol SSA and surface albedo are extremely high. Therefore, it is advisable to refrain from performing methane retrieval

over highly reflective surfaces. For aerosol SSA (0.86-0.98) and surface albedo (0.1-0.5) values commonly encountered, the
mean bias in $\Delta X_{CH_4}$ for $\Delta X_{CH_4}$-only retrieval ranges from -5.9% to 13.1% when g is fixed at 0.7. Similar to Fig.8d, Fig.8e
suggests that the retrieved $X_{alb}$ value increases with the increase in SSA.

When simultaneously retrieving methane and aerosols, Fig. 10a suggests the mean bias in $\Delta X_{CH_4}$ is significantly reduced to
0.1% by comparing with the $\Delta X_{CH_4}$-only retrieval. The STD of the retrieved methane is slightly higher when high SSA aerosols
are present over low albedo surfaces. This is explained by the near-zero AOD Jacobian values (Fig. 5b) as previously discussed.
Moreover, the STD of the retrieved $\Delta X_{CH_4}$ and AOD is a bit higher when SSA is extremely low (0.1). This decrease in retrieval
precision results from the positive values in AOD Jacobian, as well as minimal differences in AOD Jacobian between forward
and backward viewing (Fig. 5d) considering the strong absorbing characteristics of aerosols. In this scenario, it is challenging
to distinguish between aerosols and surface, thereby affecting the CH4 and aerosol retrieval. The mean bias in retrieved AOD
and $X_{alb}$ is within 1.7% and 0.07%, respectively, for typical values of aerosol SSA and surface albedo ranges (sfc alb ∈ [0.1,
0.5] and SSA ∈ [0.86, 0.98]). In general, the multi-angle viewing technique demonstrates higher accuracy compared with the
$\Delta X_{CH_4}$-only retrieval regardless of surface albedo values, especially when aerosols with stronger scattering ability are present.

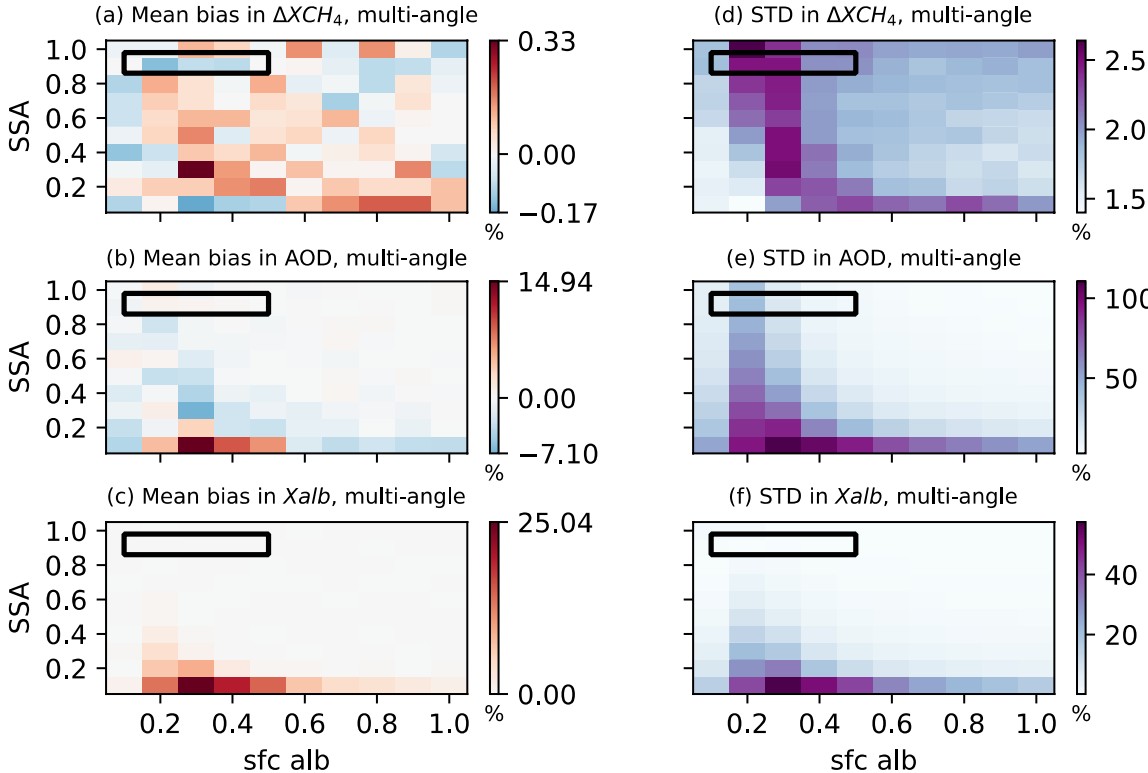

**Figure 10. Mean bias (left column) and standard deviations (STD) (right column) of retrieved $\Delta X_{CH_4}$, AOD, and $X_{alb}$
as a function of surface albedo and aerosol SSA when aerosol g is 0.7. The simulated truth of $\Delta X_{CH_4}$ and AOD are 0.1**

 **ppm and 0.1, respectively. The scattering angle ranges from 100°-140°. The black box represents the typical values for aerosol optical property and surface albedo ranges (sfc alb ∈ [0.1, 0.5] and SSA ∈ [0.86, 0.98]) in the observation.**

Apart from SSA, it is also interesting to examine how the retrieval bias varies under different combinations of aerosol asymmetry factor and surface albedo. Fig. 8c and 8f present mean bias in $\Delta X_{CH_4}$ and $X_{alb}$ for $\Delta X_{CH_4}$-only retrieval and simultaneous retrieval when aerosol SSA is fixed at 0.95. For $\Delta X_{CH_4}$-only retrieval, $\Delta X_{CH_4}$ is underestimated (overestimated) with low (high) surface albedo especially when g is small. These errors arise because aerosols with low g over dark surfaces tend to scatter more light towards the space. However, when the surface is bright, it reflects a larger proportion of light towards aerosols, and aerosols with low g tend to scatter this light back to the surface again, thereby enhancing methane absorption. The maximum bias in $\Delta X_{CH_4}$ for $\Delta X_{CH_4}$-only retrieval is around -50% when both aerosol g and surface albedo are extremely low. For the typical values of g (0.54,0.76) and surface albedo (0.1-0.5), neglecting aerosols results in a mean bias in $\Delta X_{CH_4}$ ranging from -20.5% to 12.2%.

By employing simultaneous retrieval, the mean bias in $\Delta X_{CH_4}$ can be reduced to 0.27% (Fig. 11a), demonstrating an enhancement in $\Delta X_{CH_4}$ accuracy. An increase in surface albedo enhances surface aerosol multiple scattering, while a decrease in g enhances aerosol backscattering. This competition effect results in a slope in the distribution of the large STD values. Regarding the retrieved AOD and $X_{alb}$, their mean bias falls within -4.9% and 0.06% (Fig.11b and 11c) in the presence of strongly scattering aerosols (SSA=0.95).

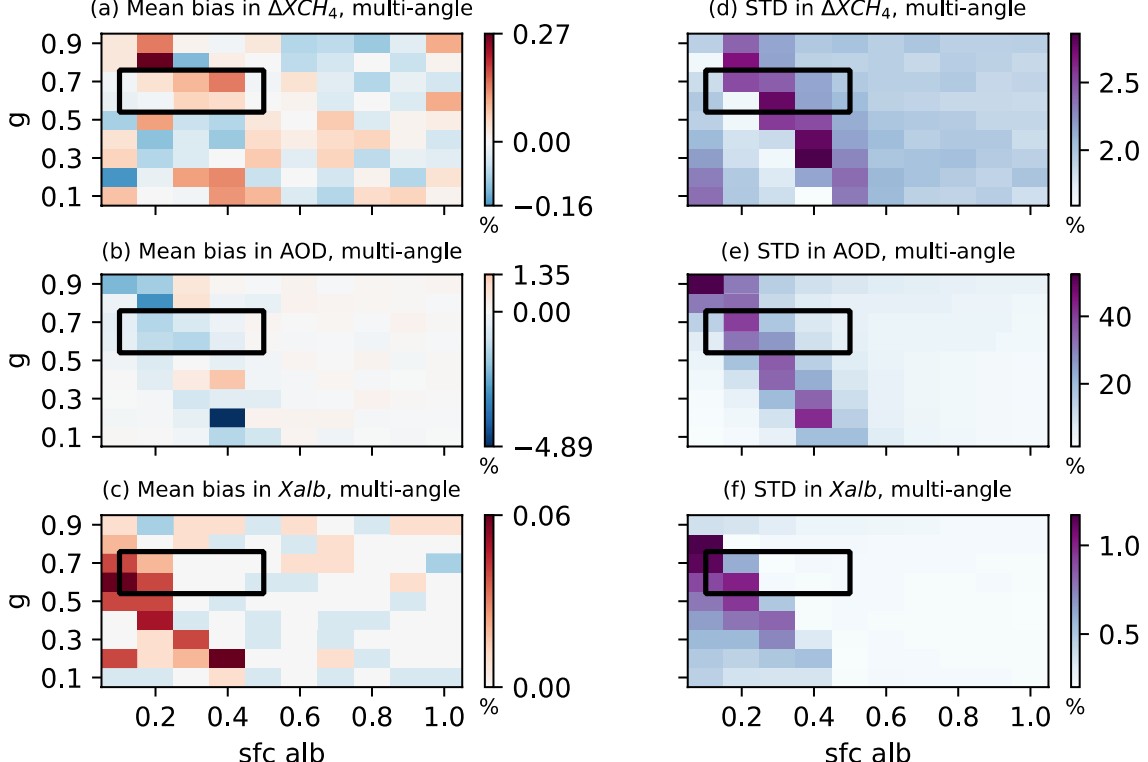

**Figure 11.** Mean bias (left column) and standard deviations (STD) (right column) of retrieved $\Delta X_{CH_4}$, AOD, and $X_{alb}$ as a function of surface albedo and aerosol g when aerosol SSA is 0.95. The simulated truth of $\Delta X_{CH_4}$ and AOD are 0.1 ppm and 0.1, respectively. The scattering angle ranges from 100°-140°. The black box represents the typical values for aerosol optical property and surface albedo ranges (sfc alb ∈ [0.1, 0.5] and g ∈ [0.54, 0.76]) in the observation.

Overall, in simultaneous $\Delta X_{CH_4}$ and AOD retrieval using the multi-angle viewing method, the retrieved values for $\Delta X_{CH_4}$, AOD, and $X_{alb}$ values generally match very well with the simulated truth across various aerosol optical properties and surface albedo conditions. Table 2 summarizes the mean bias and STD in retrieved $\Delta X_{CH_4}$, AOD, and $X_{alb}$ for the $\Delta X_{CH_4}$-only retrieval in the nadir viewing mode and simultaneous $\Delta X_{CH_4}$ and AOD retrieval in the multi-angle viewing mode, considering typical values of aerosol optical properties and surface albedo encountered in the observation. Using the simultaneous retrieval method, the mean bias and STD in $\Delta X_{CH_4}$ fall within the range of 0.3% and 2.8%, respectively. Similarly, the mean bias in AOD and $X_{alb}$ remains within 3.1% and 0.1%, respectively. It should be noted that under certain conditions characterized by near zero AOD Jacobian values, such as scenarios with high SSA and high g values over low albedo surface, and high SSA and low g values over moderately reflective surface, or positive AOD Jacobian values when SSA is extremely low over surfaces with medium to high albedo, we observe a slightly higher STD in simultaneous retrieval. Although the retrieved AOD shows relatively high accuracy, its STD can exceed 10%, suggesting the uncertainty in AOD retrieval when SSA and g are not constrained.

**Table 2** Mean bias and STD in retrieved $\Delta X_{CH_4}$, AOD, and $X_{alb}$ for the $\Delta X_{CH_4}$-only retrieval in the nadir viewing mode and simultaneous $\Delta X_{CH_4}$ and AOD retrieval in the multi-angle viewing mode with a 20° maximum satellite zenith angle. The simulated truth of $\Delta X_{CH_4}$ and AOD is 0.1 ppm and 0.1, respectively. Mean bias and STD are relative to the background values. Experiments #1 to #3 correspond to Section 3.2, and Experiment #4 corresponds to Section 4.1.

| | Mean bias in $\Delta X_{CH_4}$ | STD in $\Delta X_{CH_4}$ | Mean bias in AOD | STD in AOD | Mean bias in $X_{alb}$ | STD in $X_{alb}$ | Corr coef ($\Delta X_{CH_4}$&AOD) |
|---|---|---|---|---|---|---|---|
| *Experiment #1: SSA ∈ [0.86, 0.98], g ∈ [0.54, 0.76], sfc alb = 0.2* | | | | | | | |
| $\Delta X_{CH_4}$-only nadir retrieval | -3.0% ~ 6.3% | 1.6% | - | - | -5.7%~3.4% | 0.2% | - |
| $\Delta X_{CH_4}$ & AOD multi-angle retrieval | -0.1% ~ 0.1% | 1.6% ~ 2.7% | -1.7%~ 1.7% | 18.2%~ 48.6% | -0.07%~ 0.04% | 0.3%~ 2.1% | -85% ~ 30% |
| *Experiment #2: Sfc alb ∈ [0.1, 0.5], SSA ∈ [0.86, 0.98], g = 0.7* | | | | | | | |
| $\Delta X_{CH_4}$-only nadir retrieval | -5.9% ~ 13.1% | 1.5% ~ 1.6% | - | - | -6.7%~ 5.4% | 0.2% | - |
| $\Delta X_{CH_4}$& AOD multi-angle retrieval | -0.1% ~ 0.1% | 1.7% ~ 2.6% | - 3.1%~1.1% | 8.0%~47.0% | - 0.1%~0.04% | 0.2%~ 1.83% | -81% ~ 43% |
| *Experiment#3: Sfc alb ∈ [0.1, 0.5], g ∈ [0.54, 0.76], SSA=0.95* | | | | | | | |
| $\Delta X_{CH_4}$-only nadir retrieval | -20.5% ~ 12.2% | 1.5% ~ 1.6% | - | - | -2.3%~ 10.1% | 0.2% | - |
| $\Delta X_{CH_4}$& AOD multi- | -0.1% ~ 0.3% | 1.6% ~ 2.8% | -3.0%~ 0.7% | 4.7%~ 39.9% | 0-0.1% | 0.2%~ 1.2% | -83% ~ 52% |

| | | | | | | | |
|---|---|---|---|---|---|---|---|
| angle retrieval | | | | | | | |
| *Experiment#4: Sfc alb ∈ [0.1, 0.5], max(sat zenith) ∈ [0°,20°], SSA = 0.95, g = 0.7* | | | | | | | |
| $\Delta X_{CH_4}$-only multi-angle retrieval | -5.7%~12.4% | 1.6%~1.7% | - | - | -2.3%-5.1% | 0.2% | - |
| $\Delta X_{CH_4}$& AOD multi-angle retrievals | -0.1%~0.1% | 1.8%~2.2% | -0.2%~0.8% | 6.6%~26.8% | 0% | 0.2%~0.6% | -65%~42% |


## 4 Simultaneous Retrieval Analysis

### 4.1 The Effect of Satellite Zenith Angle on Simultaneous Retrieval

The discussions above have proved that using the multi-angle viewing method for simultaneous $\Delta X_{CH_4}$ anAOD retrievals can significantly improve the retrieval accuracy of $\Delta X_{CH_4}$ by comparing it with the $\Delta X_{CH_4}$-only nadir retrieval. It is still worth

investigating whether the retrieval results are highly dependent on the chosen satellite zenith angles. In this section, satellite zenith angles ranging from 0° to 80° are tested in both the $\Delta X_{CH_4}$-only retrieval and the simultaneous retrieval. As shown in Table 3, the scattering angle range broadens with increasing satellite zenith angle magnitude, which could benefit aerosol retrieval as it leads to more distinct differences in TOA reflectance across various satellite viewing positions. However, larger satellite zenith angles could also introduce more bias to methane retrieval because of its slant path effect.

Considering aerosols with an AOD of 0.1, a SSA of 0.95, and a g of 0.7, the mean bias and STD for the $\Delta X_{CH_4}$-only retrieval and simultaneous retrieval as a function of surface albedo and the maximum magnitude of the satellite zenith angle is shown in Fig. 12. If aerosols are neglected, the retrieved $\Delta X_{CH_4}$ are always overestimated except for the extremely low surface albedo (0.1) condition. The retrieval bias magnitude escalates with the growing maximum magnitude of the satellite zenith angle. A larger satellite zenith angle brings in a longer light path, which enhances atmospheric absorption and introduces larger retrieval

errors. The maximum mean bias in $\Delta X_{CH_4}$ for $\Delta X_{CH_4}$-only retrieval can exceed 80% when the satellite zenith angle exceeds 70°. For typical GHGSat satellite zenith angle (10°-20°) and surface albedo (0.1-0.5) ranges, the mean bias in $\Delta X_{CH_4}$ for $\Delta X_{CH_4}$-only retrieval is -5.7%~12.4%.

For simultaneous $\Delta X_{CH_4}$ and $AOD$ retrieval, the mean bias in $\Delta X_{CH_4}$ remains below 0.1% and it varies little with the chosen satellite zenith angle. This suggests that the multi-angle viewing method is effective for GHGSat-like satellites, regardless of their observation swath. The better retrieval performance of simultaneous retrieval in the multi-angle viewing mode largely results from adding AOD as an additional predictor instead of applying the multi-angle method, considering GHGSat Satellite is an intensity-only instrument targeting one specific band.

For the STD in $\Delta X_{CH_4}$ from the simultaneous retrieval, its magnitude experiences a slight increase and then decreases as the satellite zenith angle magnitude increases. This happens because, with the increase of the satellite's zenith angle, more energy scatters back to space while a longer light path leads to greater atmospheric absorption. At a specific point, the aerosol Jacobian approaches zero, which introduces relatively high uncertainty into the simultaneous retrieval process.

**Table 3 Satellite zenith angle ranges tested for $\Delta X_{CH_4}$-only retrieval and simultaneous $\Delta X_{CH_4}$ and $AOD$ retrieval using the multi-angle viewing method. The solar zenith angle is 60°.**

| Satellite zenith angle range | 0° | -10 ° ~ 10° | -20 ° ~ 20° | -30 ° ~ 30° | -40 ° ~ 40° | -50 ° ~ 50° | -60 ° ~ 60° | -70 ° ~ 70° | -80 ° ~ 80° |
|---|---|---|---|---|---|---|---|---|---|
| Scattering angle range | 120° | 110 ° ~ 130° | 100 ° ~ 140° | 90 ° ~ 150° | 80 ° ~ 160° | 70 ° ~ 170° | 60 ° ~ 180° | 50 ° ~ 180° | 40 ° ~ 180° |

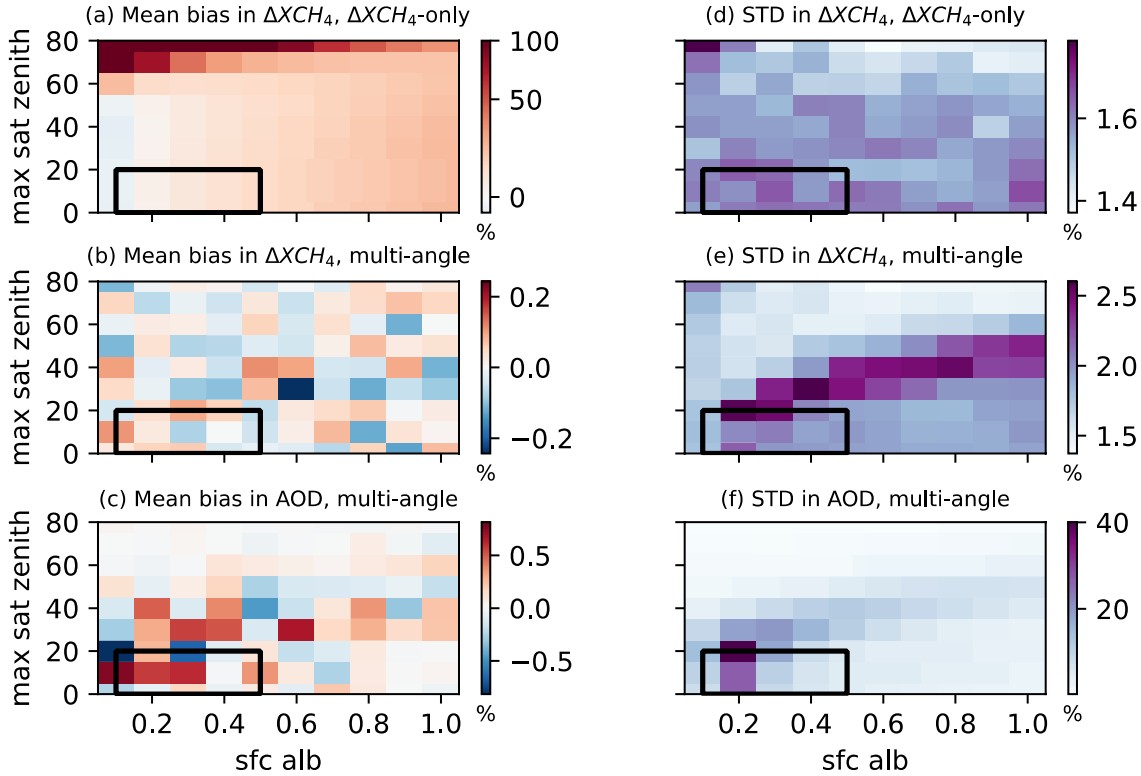


**Figure 12. (a) Mean bias and (d) standard deviations (STD) of retrieved $\Delta X_{CH_4}$ values when aerosols are present but not retrieved; (b) Mean bias and (e) STD of retrieved $\Delta X_{CH_4}$ values for simultaneous $\Delta X_{CH_4}$ and AOD retrieval; (c) Mean bias and (f) STD of retrieved $X_{AOD}$ values for simultaneous $\Delta X_{CH_4}$ and AOD retrieval; Retrieval results are displayed as a function of surface albedo and maximum magnitude of satellite zenith angle when aerosol SSA is 0.95**

**and g is 0.7, and the solar zenith angle is 60°. The satellite is in the multi-angle viewing mode. The black box represents the typical values for GHGSat satellite zenith angle and surface albedo ranges (max(sat zenith) $\theta_2 \in [0°, 20°]$ and sfc alb $\in [0.1, 0.5]$).**

## 4.2 Relationship between Retrieved $\Delta X_{CH_4}$ and $AOD$ from Simultaneous Retrieval

Fig. 13 illustrates the correlation coefficients between the retrieved $\Delta X_{CH_4}$ and AOD for various combinations of SSA, g,

surface albedo, and satellite zenith values. The simultaneous retrieval is conducted under four specific conditions using the multi-angle viewing method: (1) when the surface albedo is 0.2, (2) when the g is 0.7, (3) when the SSA is 0.95, (4) when the SSA is 0.95 and g is 0.7. For conditions (1) to (3), the angle setting follows Table 1, while for condition (4), the angle settings are based on Table 3.

Fig. 13a suggests that $\Delta X_{CH_4}$ and AOD are negatively correlated for high g values and negatively correlated for low g values

when the surface is dark. A high g results in more concentrated forward scattering towards the ground, causing more atmospheric absorption via aerosol-surface multiple scattering. To maintain the relative depth of the CH4 absorption spectra,

less $\Delta X_{CH_4}$ needs to be retrieved to balance the effect of increasing AOD. In Fig. 13b, $\Delta X_{CH_4}$ and AOD are positively correlated for low albedo surfaces and negatively correlated for mid and high albedo surfaces when g is 0.7. With a dark surface, the increase of aerosols scatters a greater amount of light back to space, leaving less light to interact with $CH_4$. Consequently, a larger $\Delta X_{CH_4}$ is retrieved to counterbalance the impact of increasing AOD. Fig. 13c shows that the correlation between $\Delta X_{CH_4}$ and AOD changes from positive to negative with the increase of g and surface albedo when SSA is 0.95. This pattern occurs because of the shift in dominant aerosol-involved physical processes from the aerosol-only scattering effect to the aerosol and surface multiple scattering effects. Fig. 13d shows that for aerosols with an SSA of 0.95 and a g of 0.7, $\Delta X_{CH_4}$ and AOD are positively (negatively) correlated at low(high) albedo. With the increase of the satellite zenith angle, the magnitude of the correlation coefficient first increases and then decreases, suggesting that it is still beneficial to apply large scattering angle ranges in the multi-angle viewing method to better distinguish aerosols and methane.

When considering a surface with an albedo of 0.2, a SSA from 0.86 to 0.98, and a g from 0.54 to 0.76, the correlation coefficient between the retrieved $\Delta X_{CH_4}$ and AOD falls within the range of -85% to 30%. Similarly, when the SSA is maintained between 0.86 and 0.98, the surface albedo varies from 0.1 to 0.5, and g is fixed at 0.7, the correlation coefficient ranges from -81% to 43%. Lastly, for cases where g ranges from 0.54 to 0.76, the surface albedo spans from 0.1 to 0.5, and SSA is set at 0.95, the correlation coefficient varies from -83% to 52%. In general, The pattern in Fig.13 is similar to the $\Delta X_{CH_4}$ STD pattern in Fig. 9-12, which confirms that the highly correlated $\Delta X_{CH_4}$ and AOD results in larger STD in $\Delta X_{CH_4}$.

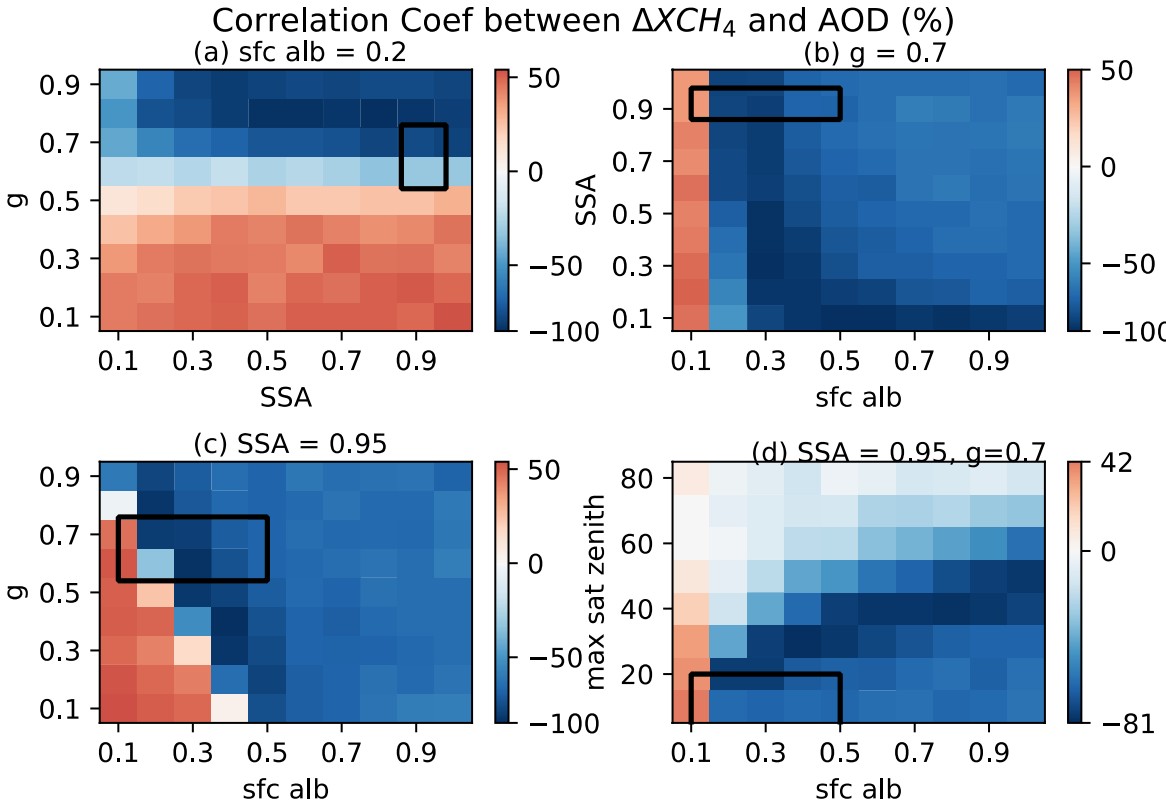

**Figure 13.** Correlation coefficient (%) between simultaneously retrieved methane enhancement ($\Delta X_{CH_4}$) and aerosol optical depth (AOD) under varying aerosol types and surface albedo values. (a) fix surface albedo as 0.2; (b) fix aerosol g as 0.7; (c) fix aerosol SSA as 0.95. For (a)-(c), the maximum magnitude of the satellite zenith angle is 20°. (d) fix aerosol SSA as 0.95 and g as 0.7. The black box represents the typical values for aerosol optical property, surface albedo, and solar zenith angle ranges in the GHGSat observation.

### 4.3 Impact of Aerosol and Surface Albedo Uncertainties on Simultaneous Retrieval

Although aerosol types could be inferred from emission plumes by considering the combustion type and its location, the uncertainty that arises from inaccurate representation of aerosol types and distributions could impact the performance of our simultaneous retrieval. Additionally, assumptions regarding the Lambertian surface and satellite viewing geometry could potentially introduce uncertainties in surface albedo retrieval. To access such uncertainty, we employ certain aerosol SSA and g, height distributions, and surface albedo in retrieval, while for the simulated GHGSat radiance, we incorporate more complex representations of aerosol type and distributions, and surface albedo. The differences between retrieval with fixed (inaccurate) parameters and retrieval with real (accurate) parameters enable us to quantify the uncertainty resulting from the inaccurate representation of these parameters.

### 4.3.1 Aerosol Type Uncertainties

Fig. 14 presents the differences in mean bias and standard deviations of retrieved variables between retrieval assuming SSA =
0.95 and g = 0.7 for aerosols and retrieval assuming the correct SSA and g (ranging from 0 to 1). These differences could
suggest the uncertainty of simultaneous retrieval when assuming inaccurate aerosol types. Fig.14a and 14d show that the
uncertainty in the mean bias and STD of $\Delta X_{CH_4}$ related to aerosol types ranges from -5.8% to 2.7% and -0.2 to 0.9%,
respectively for typical aerosol optical property values. The uncertainty in the mean bias and STD of AOD falls within -40.2%
to 16.1% and -9.6% to 20%, respectively. Similarly, the uncertainty in the mean bias and STD of $X_{alb}$ ranges from -5.6% to
5.4% and -1.5% to 0.39%, respectively. These findings suggest that even with incorrect SSA and g assumptions in the retrieval,
the maximum uncertainty induced in the accuracy of retrieved $\Delta X_{CH_4}$ is within 5.8%.

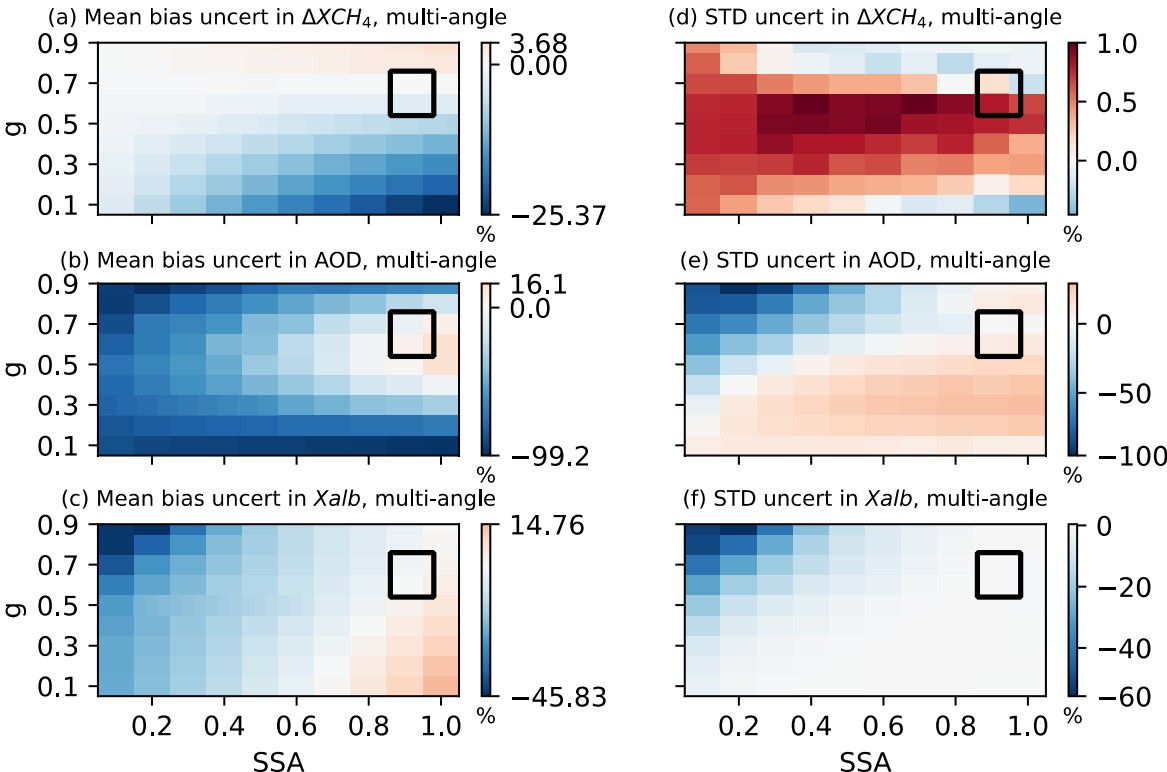

**Figure 14. Uncertainties induced by aerosol type in mean bias (left column) and standard deviations (STD) (right
column) of retrieved $\Delta X_{CH_4}$, AOD, and $X_{alb}$, assuming aerosols with an SSA of 0.95 and a g of 0.7 in the retrieval. The**
**simulated truth of $\Delta X_{CH_4}$, AOD, and $X_{alb}$ are 0.1 ppm, 0.1, and 0.2, respectively. The scattering angle ranges from 100°-
140°. The black box represents the typical values for aerosol optical property ranges (SSA ∈ [0.86, 0.98] and g ∈ [0.54,
0.76]) in the observation.**

### 4.3.2 Aerosol Height Distribution Uncertainties

While aerosols primarily reside near the surface at the industrial site, they could also ascend to higher altitudes under favorable
atmospheric conditions. Therefore, we examined the uncertainty brought by aerosol height assumptions. We compared the
differences between the retrieval when we assume aerosols are near the surface with those when aerosols are elevated to 5 km.
In the latter case, AOD linearly decreases with height but we still use the near-surface Jacobian calculations in retrieval. Fig.15
shows the uncertainties in simultaneous retrieval when assuming incomplete aerosol height.

Similar to the uncertainty results related to aerosol types, Fig.15a and 15d show that the uncertainty induced by aerosol height
in the mean bias and STD of $\Delta X_{CH_4}$ ranges from 2.3% to 6.4% and from -0.1 to 0.1%, respectively, for typical values of aerosol
optical properties. The mean bias uncertainty for AOD and $X_{alb}$ falls within the range of 2.3% to 41.5% and -0.8 to 1.4%,
respectively. The STD uncertainty for $\Delta X_{CH_4}$, AOD, and $X_{alb}$ is generally small, indicating minimal sensitivity of retrieval
precision to the aerosol height distributions.

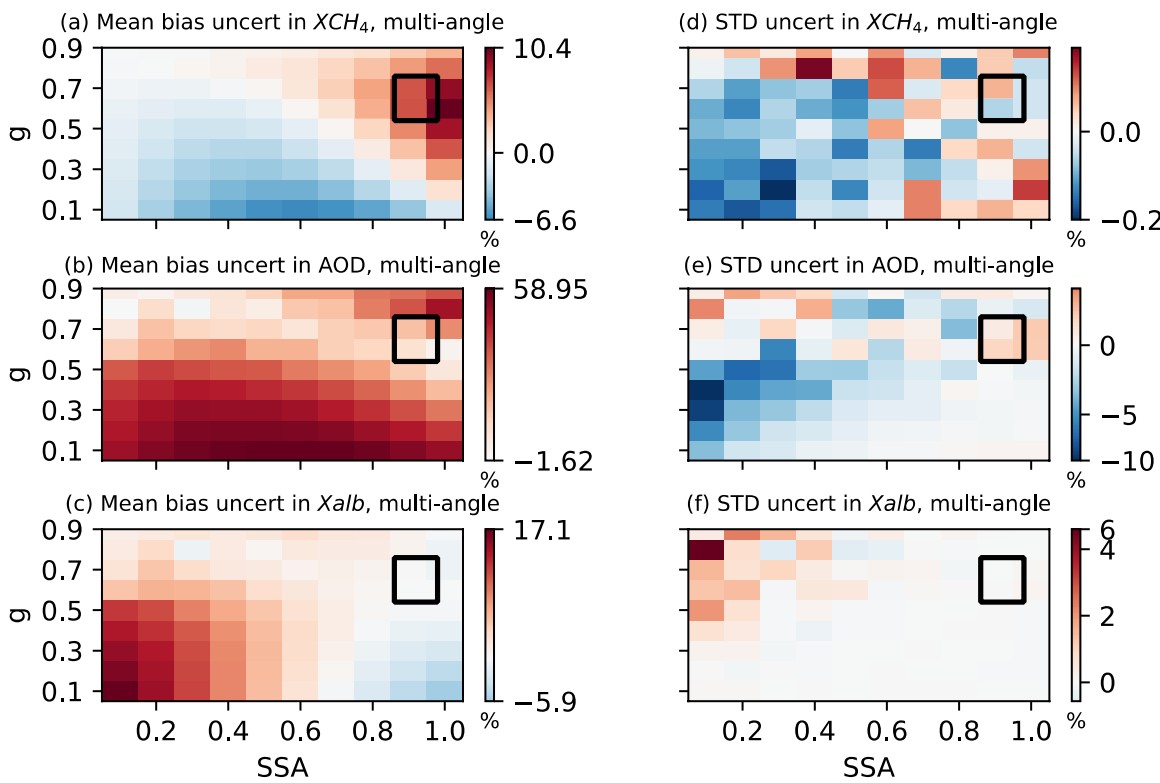

**Figure 15. Uncertainties induced by aerosol height distributions in mean bias (left column) and standard deviations
(STD) (right column) of retrieved $\Delta X_{CH_4}$, AOD, and $X_{alb}$, assuming near-surface aerosols in retrieval. The simulated
truth of $\Delta X_{CH_4}$, AOD, and $X_{alb}$ are 0.1 ppm, 0.1, and 0.2, respectively. The scattering angle ranges from 100°-140°. The
black box represents the typical values for aerosol optical property ranges (SSA ∈ [0.86, 0.98] and g ∈ [0.54, 0.76]) in
the observation.**

### 4.3.3 Surface Albedo Uncertainties

Although a second-order polynomial has been applied in the retrieval to account for the bidirectional distribution of surface albedo, the imperfect representation of surface albedo, particularly in regions with heterogeneous landscapes, could introduce uncertainty in the simultaneous retrieval. To quantify such uncertainty, we compared the differences between the retrieval when we assume surface albedo is 0.2 with those with correct surface albedo values. Fig.16 shows the uncertainties in simultaneous retrieval when assuming imperfect surface albedo.

Fig.16a and 16d show that the uncertainty resulting from surface albedo variations in the mean bias and STD of $\Delta X_{CH_4}$ ranges from -15.1% to 4% and from -0.1 to 0.7%, respectively, for typical aerosol SSA and surface albedo ranges (sfc alb $\in$ [0.1, 0.5] and SSA $\in$ [0.86, 0.98]). The mean bias uncertainty for AOD and $X_{alb}$ falls within the range of -12.7% to 37.6% and -5.9 to 3.5%, respectively, while the STD uncertainty for AOD and $X_{alb}$ ranges from -1.1% to 31.9% and from -0.31% to 2.25%, respectively.

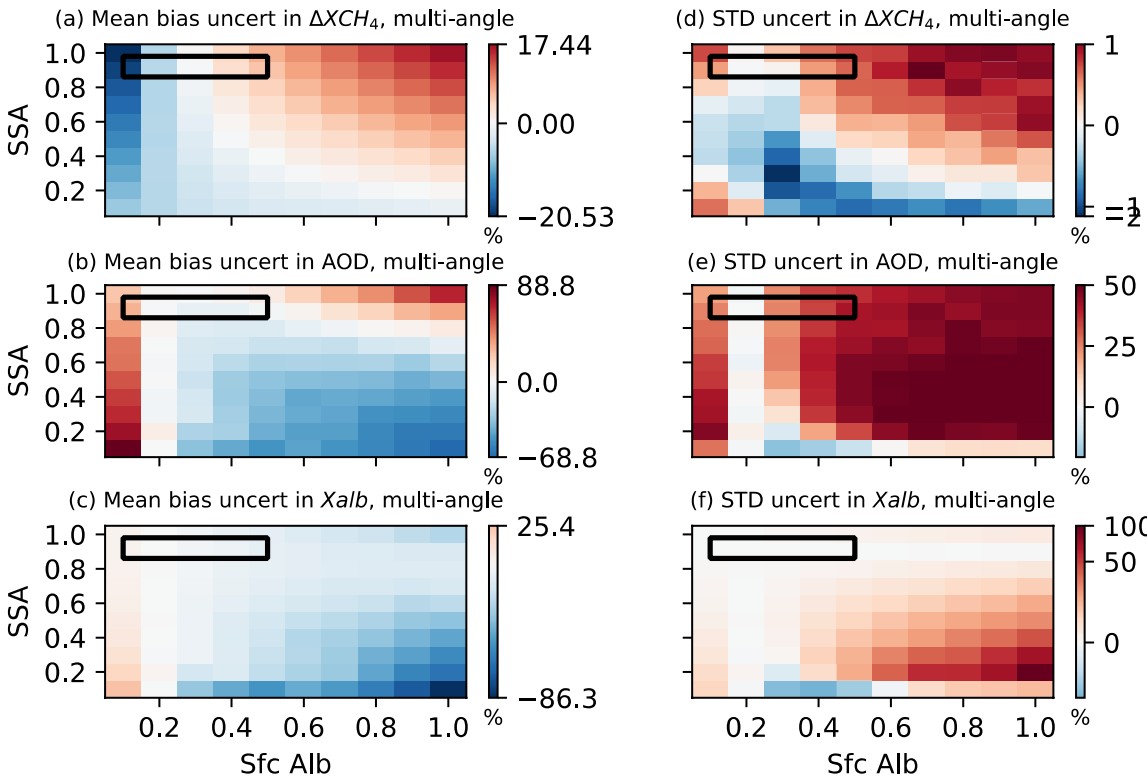

**Figure 16. Uncertainties induced by surface albedo in mean bias (left column) and standard deviations (STD) (right column) of retrieved $\Delta X_{CH_4}$, AOD, and $X_{alb}$, assuming 0.2 surface albedo in retrieval. The simulated truth of $\Delta X_{CH_4}$ and AOD are 0.1 ppm and 0.1, respectively. The scattering angle ranges from 100°-140°. The black box represents the typical values for aerosol optical property and surface albedo ranges (sfc alb $\in$ [0.1, 0.5] and SSA $\in$ [0.86, 0.98]) in the observation.**

In summary, the uncertainty in the mean bias and STD of $\Delta X_{CH_4}$ induced by inaccurate aerosol types, height distributions, and surface albedo is less than 15.1% and 0.9%, respectively. This uncertainty is obtained when assuming near-surface aerosols with fixed SSA (0.95) and g (0.7) and a 0.2 surface albedo in retrieval, while in simulated radiance, aerosol SSA, g, height distribution, and surface albedo vary across typical observation ranges.

## 5 Conclusions

This study investigates the impacts of aerosols on GHGSat methane retrieval in the shortwave near-infrared band by exploiting the dynamic aerosol scattering behaviour during the GHGSat "multi-angle" observation sequence. Specifically, this research assesses how reliably aerosols could be simultaneously retrieved with methane using the multi-angle viewing method under different aerosol optical properties, surface albedo, and satellite zenith angle conditions. Observing system simulation experiments (OSSE) are conducted to simulate GHGSat observations and perform retrieval in the presence of white noise and 1/f errors. These experiments involve a comparative assessment of retrieval accuracy and precision under two conditions: (1) when aerosols are present but not retrieved in the satellite nadir viewing mode and (2) when both methane mixing ratio enhancement ($\Delta X_{CH_4}$) and aerosol optical depth (AOD) are retrieved simultaneously in the multi-angle viewing mode.

$\Delta X_{CH_4}$-only retrieval experiment indicates the general behaviour that $\Delta X_{CH_4}$ is underestimated for low albedo surfaces and overestimated for high albedo surfaces when aerosols are not taken into account. The estimated errors in $\Delta X_{CH_4}$ for non-aerosol retrieval become more significant as aerosol single scattering albedo (SSA) increases and asymmetry factor (g) decreases. For nadir viewing simulations where AOD is set at 0.1 and the solar zenith angle at 60°, the mean bias in retrieved $\Delta X_{CH_4}$ is most significant when scattering aerosols are neglected over bright surfaces. For a surface with a 0.2 albedo, the bias in $\Delta X_{CH_4}$ varies from -3.0% to 6.3% for typical aerosol optical properties (SSA $\in$ [0.86,0.98] and g $\in$ [0.54,0.76]) (Fig. 9a); For satellite zenith angle ranging from 0°-20° and surface albedo varying between 0.1-0.5, the mean bias in $\Delta X_{CH_4}$ for $\Delta X_{CH_4}$-only retrieval spans from -5.7%~12.4% (Fig. 12a) assuming an AOD of 0.1, SSA of 0.95 and a g value of 0.7.

Using the multi-angle viewing method for simultaneous $\Delta X_{CH_4}$ and AOD retrieval, we find bias in retrieved $\Delta X_{CH_4}$ is significantly reduced at the modest cost of slightly worse $\Delta X_{CH_4}$ precision. Through simultaneous retrieval, the mean bias in $\Delta X_{CH_4}$ can be reduced to as low as 0.3% for the typical range of aerosol optical properties, surface albedo, and satellite zenith angles (Table 2). The standard deviation (STD) of $\Delta X_{CH_4}$ in simultaneous retrieval experiences a slight increase when aerosols have minimum impact on the TOA radiance, indicated by near-zero AOD Jacobian values. Nevertheless, this STD remains within a 2.8% range. The uncertainty in the mean bias and STD of $\Delta X_{CH_4}$ induced by inaccurate aerosol types, height distributions, and surface albedo is less than 15.1% and 0.9%, respectively (Fig.14 -16). The multi-angle viewing method also performs relatively well in AOD retrieval, characterized by a mean bias of less than 3.1% (Table 2). The performance assessment shows that retrieving aerosols and methane simultaneously using the multi-angle viewing method is a viable approach for operational application to GHGSat.

The correlation coefficient between simultaneously retrieved AOD and $\Delta X_{CH_4}$ switches from positive to negative with the increase of surface albedo and a decrease of aerosol g (Fig. 13a-c). This transition occurs because the dominant influence of aerosols on radiance shifts from the aerosol-only scattering effect to the aerosol-surface multiple scattering effect, which suggests that the ability to differentiate between aerosols and methane is highly dependent on the aerosols and surface conditions.

This study also explored whether the success of the AOD and $\Delta X_{CH_4}$ co-retrieval with multi-angle viewing techniques is largely determined by the range of scattering angles present in the GHGSat observation sequence. After conducting retrieval over a range of satellite zenith angle values (0° to 80°), results suggest that a broader scattering angle range, such as larger satellite zenith angle, has little impact on the improvement of AOD and $\Delta X_{CH_4}$ co-retrieval accuracy and precision. Therefore, the multi-angle viewing method is relatively insensitive to the satellite angle setting for the GHGsat-like instrument when AOD is incorporated in retrieval.

Finally, future work on the production GHGSat retrieval algorithm and real retrieval test will investigate the feasibility of adding an aerosol retrieval capability for current and future instruments.

**Data Availability**

The atmospheric model, synthetic data used by the assessment can be obtained from the Mendeley Data https://data.mendeley.com/datasets/jxcmc63p2h/1

**Author Contribution**

QY, DJ, and YH co-designed the OSSEs. QY and YH developed the radiative transfer model and DJ provided the GHGSat instrument model. QY led the writing of the manuscript with contributions from DJ and YH.

**Competing Interests**

The authors declare that they have no conflict of interest.

**Acknowledgments**

We acknowledge funding provided by the Mitacs Accelerate program (IT16447) and the Natural Sciences and Engineering Research Council of Canada (Grant RGPIN-2019-04511) that supported this research.

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
