# Peer review of "Accounting for Aerosols Effect in GHGSat Methane Retrieval"

_EGUsphere, 2023_

## Author Comment (AC1)

**Response to Reviewer Comments**

We truly appreciate the reviewer#3 for the thoughtful and helpful comments. Below are our responses (in regular font) to the reviewer's comments (in ***bolded italic*** font).

*Reviewer #3:*
***The manuscript investigates the retrieval of GHGSat-like methane retrievals in the presence of a simulated aerosol layer. Performance of the retrieval is investigated in four different configurations: $\Delta XCH4$ only in nadir and multi-angle viewing methods, and simultaneous $\Delta XCH4$ and $AOD$ in both viewing approaches. Bias and standard deviation in the retrieval are investigated under varying satellite viewing angles, aerosol SSA, asymmetry factor (g), and surface albedo. Some aspects of the work do feel simplistic, as described by Referee #1. However, the results are clearly described, and the interaction with aerosols is important to understand in the satellite methane community. I believe the work is suitable for publication, if the following comments are addressed:***

***Regarding the notation "$X_{AOD}$" used in the text, this choice seems unusual. The numerical plots already simply use "AOD", which I would encourage.***

- $X_{AOD}$ has been replaced as AOD in the manuscript.

***L9 'polarity of $\Delta XCH4$' – suggest rephrasing as the sign of $\Delta XCH4$ biases***

- Corrected.

***L104 "In Fig. 1b, strong $CH_4$ absorptions are found around 1666 nm, affirming that the DISORT-simulated radiance is adequate for simulating the methane effect." This conclusion is of importance but seems a little weak. Can it be strengthened with more evidence, or past validation?***

- GHGSat measures methane concentrations by analyzing the spectrally decomposed solar backscattered radiation within the methane absorption band ($\sim 1.65 \mu$m) (Jervis et al., 2021). If DISORT can simulate radiance identical to GHGSat and capture the methane spectra absorption features when given specific atmospheric profiles and $CH_4$ mixing ratio, it can effectively mimic the GHGSat measurement. The methane absorption features observed in TOA reflectance, as demonstrated in our paper in Figure 1b, align closely with results from other studies, such as Figure 3a in (Jervis et al., 2021) and Figure 2 in (Chan Miller et al., 2023). This consistency suggests the robustness and validity of our radiance simulations. In our study, we combine the LBLRTM, DISORT, and GHGSat instrument models as the

forward model to simulate GHGSat measurements. Following the reviewer's suggestion, discussions have been added at Lines 119-122.

*L134 "solar and satellite zenith angles" is repeated twice. The second occurrence should be azimuth*

- Corrected.

*Fig 2 It would be preferable to draw the phi2 angle to the projected point (possibly lying on the x-axis), not to the dashed backward viewing line. Often, these figures are labeled with North/South. Additionally, I am now realizing that this figure does not show the same set of angles as Table 1, which seems a little confusing, though not necessarily 'wrong' since it is a schematic diagram.*

- Thank you for the suggestions. Figure 2 has been redrawn completely.

*I would like to understand the difference between Fig 3 and Fig 1(c). The main two differences seem to be: the 0.3 FWHM smoothing and the added simulation of the 0.1 AOD aerosol layer. Looking at regions such as 1664-1665 nm, I do not think the coarser resolution explains the quantitative differences of values near ~0.2 compared to ~0.15. Is this then explained by the aerosol layer, or also other factors?*

- Thank you for the comment. In Figure 3, we added sulfate aerosols with an AOD of 0.1 in the DISORT simulation. The results in Figure 3 have been updated. The differences in simulated reflectance between Figure 3 (green line, nadir view) and Figure 1(c) are purely attributed to aerosols. As shown below in Fig.S1, we have plotted the simulated GHGSat TOA reflectance for scenarios of clean atmospheric conditions and AOD 0.1 for comparison. Sulfate aerosols introduce more atmospheric scattering and result in a slightly higher TOA reflectance. The magnitude of aerosol-induced TOA reflectance change is around $10^{-3}$. Relevant discussions are added at Lines 197-200.

[Figure]

*Figure S3. Upper: Simulated TOA reflectance measured by GHGSat instrument at a spectral resolution of 0.3 nm FWHM for clean condition and AOD = 0.1 condition. Bottom: Reflectance differences between AOD =0.1 condition and clean condition. The instrument observes the surface with an albedo of 0.2 at nadir viewing positions. For the AOD condition, sulfate aerosols with 0.1 AOD at SWIR are added near the surface.*

**In the title of Figure 3, what is SU in AOD(SU)?**

- SU stands for sulfate aerosols. The title and caption of Figure 3 have been changed accordingly.

**In Figure 4, can the first step (labeled DISORT), be a little descriptive, to distinguish it from DISORT in the forward model step?**

- Thank you for the suggestion. Figure 4 has been updated.

**L216-217 Note that Fig 6b (phase function for specific g) is not explicitly mentioned, and the link could be clearer to what is discussed (intensity of scattering energy)**

- Descriptions of Fig 6b have been added at Lines 276-279.

**L229 I do not necessarily understand the uncertainty simulation here. Is 0.2% the combined magnitude of the white noise and 1/f noise, or for each individually? Is uncertainty of the aerosol optical properties included somehow?**

- In response to the reviewer's comment about the 0.2% magnitude of noise, we introduced white noise and 1/f errors, each with a standard deviation of 0.2%, to account for instrument measurement uncertainty. These settings are considered reasonable within the GHGSat system. Clarifications are added at Line 298.

- Regarding the reviewer's comment on the uncertainties introduced by aerosol optical properties, we have added results in Section 3.2. We assumed certain aerosol SSA, g, and height distributions in retrieval (e.g. Jacobian calculation), while for the simulated GHGSat radiance, we incorporated more complex representations for aerosol type and height distributions. The differences between retrieval with fixed (inaccurate) parameters and retrieval with real (accurate) parameters enable us to quantify the uncertainty resulting from the inaccurate representation of these parameters.

**Aerosol Type Uncertainties**

Fig. S2 presents the differences in mean bias and standard deviations of retrieved variables between retrievals assuming SSA = 0.95 and g = 0.7 for aerosols and retrievals assuming the correct SSA and g. These differences could suggest the uncertainty of simultaneous retrieval when assuming inaccurate aerosol types. Fig.S2a and S2d show that the uncertainty in the mean bias and STD of $\Delta X_{CH_4}$ related to aerosol types ranges from -5.8% to 2.7% and -0.2 to 0.9%, respectively, for typical aerosol optical property values. The uncertainty in the mean bias and STD of AOD falls within -40.2% to 16.1% and -9.6% to 20%, respectively. Similarly, the uncertainty in the mean bias and STD of $X_{alb}$ ranges from -5.6% to 5.4% and -1.5% to 0.39%, respectively. These findings suggest that even with incorrect SSA and g assumptions in the retrieval, the maximum uncertainty induced in the accuracy of retrieved $\Delta X_{CH_4}$ is within 5.8%.

[Figure]

*Figure S2. Uncertainties induced by aerosol type in mean bias (left column) and standard deviations (STD) (right column) of retrieved $\Delta X_{CH_4}$, AOD, and $X_{alb}$, assuming aerosols with an SSA of 0.95 and a g of 0.7 in the retrieval. The simulated truth of $\Delta X_{CH_4}$, AOD, and $X_{alb}$ are 0.1 ppm, 0.1, and 0.2, respectively. The scattering angle ranges from 100°-140°. The black box represents the typical values for aerosol optical property ranges (SSA ∈ [0.86, 0.98] and g ∈ [0.54, 0.76]) in the observation.*

**Aerosol Height Distribution Uncertainties**

While aerosols primarily reside near the surface at the industrial site, they could also ascend to higher altitudes under favorable atmospheric conditions. Therefore, we examined the uncertainty brought by aerosol height assumptions. We compared the differences between the retrieval when we assume aerosols are near the surface with those when aerosols are elevated to 5 km. In the latter case, AOD linearly decreases with height but we still use the near-surface Jacobian calculations in retrieval. Fig.S3 shows the uncertainties in simultaneous retrieval when assuming incomplete aerosol height.

Similar to the uncertainty results related to aerosol types, Fig.S3a and S3d show that the uncertainty induced by aerosol height in the mean bias and STD of $\Delta X_{CH_4}$ ranges from 2.3% to 6.4% and from -0.1 to 0.1%, respectively, for typical values of aerosol optical

properties. The mean bias uncertainty for AOD and $X_{alb}$ falls within the range of 2.3% to 41.5% and -0.8 to 1.4%, respectively. The STD uncertainty for $\Delta X_{CH_4}$, AOD, and $X_{alb}$ is generally small, indicating minimal sensitivity of retrieval precision to the aerosol height distributions.

[Figure]

*Figure S3. Uncertainties induced by aerosol height distributions in mean bias (left column) and standard deviations (STD) (right column) of retrieved $\Delta X_{CH_4}$, AOD, and $X_{alb}$, assuming near-surface aerosols in the retrieval. The simulated truth of $\Delta X_{CH_4}$, AOD, and $X_{alb}$ are 0.1 ppm, 0.1, and 0.2, respectively. The scattering angle ranges from 100°-140°. The black box represents the typical values for aerosol optical property ranges (SSA ∈ [0.86, 0.98] and g ∈ [0.54, 0.76]) in the observation.*

In summary, the uncertainty in the mean bias and STD of $\Delta X_{CH_4}$ induced by inaccurate aerosol types and height distributions is less than 6.4% and 0.9%, respectively. This uncertainty is obtained when assuming near-surface aerosols with fixed SSA (0.95) and g (0.7) and a 0.2 surface albedo in retrieval, while in simulated radiance, aerosol SSA, g, and height distribution vary across typical observation ranges.

*L356 'A bunch of' – Suggest to quantitatively specific the satellite zenith angles considered*

- Corrected.

*While satellite zenith angles were considered, what about solar zenith angle? I believe this was fixed to 60 degrees throughout and wonder how this influences the results.*

- Adjusting the solar zenith angle gives us retrieval results similar to those discussed in section 4.1. When we alter the solar and satellite zenith angles, the scattering angle changes, thereby influencing our retrieval. In section 4.1, we have examined extremely wide scattering angle ranges (40° ~ 180°), which are likely among the most extreme values seen in real observations. Figure 12 indicates that simultaneous methane and aerosol retrievals exhibit relatively small mean bias when the satellite zenith angle (scattering angle range) is small. This suggests that there is little requirement for the angle setting when applying the multi-angle viewing method to the GHGSat instrument.

*Figure 8 and others, note there is a discrepancy between the figure ("Mean bias in XCH4") and caption (Mean bias of ΔXCH4)*

- Thank you for the comments. All captions are updated accordingly.

**References**

Chan Miller, C., Roche, S., Wilzewski, J. S., Liu, X., Chance, K., Souri, A. H., Conway, E., Luo, B., Samra, J., Hawthorne, J., Sun, K., Staebell, C., Chulakadabba, A., Sargent, M., Benmergui, J. S., Franklin, J. E., Daube, B. C., Li, Y., Laughner, J. L., Baier, B. C., Gautam, R., Omara, M., and Wofsy, S. C.: Methane retrieval from MethaneAIR using the $CO_2$ Proxy Approach: A demonstration for the upcoming MethaneSAT mission, Gases/Remote Sensing/Data Processing and Information Retrieval, https://doi.org/10.5194/egusphere-2023-1962, 2023.

Jervis, D., McKeever, J., Durak, B. O. A., Sloan, J. J., Gains, D., Varon, D. J., Ramier, A., Strupler, M., and Tarrant, E.: The GHGSat-D imaging spectrometer, Atmos. Meas. Tech., 14, 2127–2140, https://doi.org/10.5194/amt-14-2127-2021, 2021.

---

## Author Comment (AC2)

**Response to Reviewer Comments**

We truly appreciate reviewer#1 for the thoughtful and helpful comments. Below are our responses (in regular font) to the reviewer's comments (in ***bolded italic*** font).

*Reviewer #1:*
*In this paper, numerical investigations are carried out on the effects of aerosol retrievals on methane remote sensing with GHGSat. The authors first used a radiative transfer model and the multi-angle viewing method to generate "GHGSat measurements", then the aerosols and methane were retrieved using a retrieval method. If the retrieval method is just right, we can clearly see how the aerosols affect the methane retrieval as there are small uncertainties in the "measurements". This work provides valuable information for methane measurements with the new instrument, so this study certainly falls within the scope of AMT and the results could be of great importance to the scientific community.*

*However, as shown in my initial review, the authors should first clearly review and discuss the applicability of the interrogation method, as this would significantly affect the conclusion. The specific comments are listed below:*

*(1) The authors try to show that the retrieval of aerosols is important for methane retrieval. However, in the simulations, many parameters were fixed in the determination of aerosols. In real cases, these parameters are generally unknown (e.g. surface albedo). Therefore, the retrieval of aerosols in real measurements may lead to larger uncertainties in the methane remote sensing than shown in this manuscript.*

- In response to the reviewer's comment about 'many parameters being fixed in the determination of aerosols', we acknowledge that we fixed several variables in the retrieval experiments. This is done to better understand the impact of non-fixed parameters and their associated uncertainties on the simultaneous retrieval process. Many factors can influence the simultaneous retrieval of methane and aerosols, including aerosol optical properties, surface albedo, and satellite viewing angles. To isolate the effect of each factor, we keep the remaining factors constant to examine how the accuracy and precision of retrieval change with all possible combinations of these factors. For example, to assess the effect of aerosol single scattering albedo (SSA) and asymmetry factor (g) when we only retrieve AOD for the aerosol-related parameter, we assume the background surface albedo is 0.2 and examine how the mean bias and STD vary with different combinations of aerosol SSA and g. Fig.9 suggests that even with extreme and unrealistic SSA and g values, the simultaneous retrieval can maintain the mean bias and standard deviation (STD) of $\Delta X_{CH_4}$ within 0.15% and 2.5%, respectively. For typical values of aerosol SSA and g ranges in

the observation, the mean bias in retrieved AOD and $X_{alb}$ are within 1.7% and 0.07%, respectively. (Fig.9 and Table 2), indicating robust performance of AOD and $X_{alb}$ retrieval across diverse aerosol-type conditions. Similarly, to account for the interaction between aerosols and surface albedo, we further fix g at 0.7 (or SSA at 0.95) to investigate how mean bias and STD change with SSA (or g) and surface albedo. These fixed values (surface albedo 0.2; SSA 0.95; g 0.7) are chosen as they are typical in the observations. Our results (Fig.10 and Fig.11) demonstrate mean bias and STD of $\Delta X_{CH_4}$ within 0.3% and 2.5% regardless of surface albedo and SSA or g values. These findings suggest that simultaneous $\Delta X_{CH_4}$ and AOD retrieval performs well across various aerosols and surface simulation configurations. Therefore, although we fixed certain variables in the retrieval for demonstration, the retrieval results are still robust across different retrieval configurations. Explanations regarding the reasons for fixing parameters are included in Lines 346-348.

- In response to the reviewer's comment that 'many parameters are generally unknown (e.g. surface albedo)', it should be noted that surface albedo is consistently retrieved in our study. In $\Delta X_{CH_4}$-only retrieval scenario, we obtain both surface albedo ($X_{alb}$) and methane mixing ratio ($\Delta X_{CH_4}$). In simultaneous $\Delta X_{CH_4}$ and AOD retrieval scenario, we retrieve $X_{alb}$, AOD, and $\Delta X_{CH_4}$. To account for the bidirectional distribution of surface albedo and the per-pixel signal changes resulting from satellite motion, we also include a second-order polynomial as a function of the image frame index n in the forward model (Jervis et al., 2021) as shown in Eq. (5) and (7). Clarifications are added in Figure 4 and Lines 217-218 and 223. The $X_{alb}$ retrieval results are added in Figs 8-11.

- Regarding the reviewer's comment on the potential uncertainties introduced by real aerosol measurements in methane remote sensing, we appreciate this comment and agree that idealizing aerosols (e.g. height distribution) and inaccurate representations of aerosol types and surface albedo in reality could potentially impact the accuracy of simultaneous retrieval. To access such uncertainty, we assume certain aerosol SSA, g, height distributions, and surface albedo in retrieval (e.g. Jacobian calculation), while for the simulated GHGSat radiance, we incorporate more complex representations for aerosol type, height distributions, and surface albedo. The differences between retrieval with fixed

[revised manuscript text omitted]

ranges. To retrieve more aerosol information and further reduce uncertainty, additional constraints such as multi-band observations are required. However, implementing such measures is not feasible at the moment as the current GHGSat instrument only targets the 1.65 μm band. Discussions about retrieval uncertainty are added in Section 4.3.

*(2) As my first review has shown, the authors should go into more detail on the applicability of the retrieval algorithm. Is the retrieval algorithm correct for the real cases? If it is not correct, the improvement in retrieval accuracy when adding aerosol retrieval does not necessarily mean that aerosol retrieval is important, and it may also be because the algorithm is wrong. Of course, the authors used their retrieval accuracy to show the applicability of the algorithm. However, their simulations and retrievals were simplified by many fixed parameters and assumptions. I suggest that the authors perform a sensitivity analysis for more complex parameters. If the authors can prove that the retrieval is accurate in different and more complex simulation configurations, the retrieval algorithm is credible.*

Thank you for the insightful comments.

- In response to the reviewer's comment about 'the applicability of the retrieval algorithm', the simultaneous $\Delta X_{CH_4}$ and AOD retrieval method is primarily advantageous in two aspects: it enhances the methane gas retrieval accuracy by accounting for aerosols effect for GHGSat-like point source imagers and it measures aerosol plumes using such imagers. In this study, the integration of LBLRTM, DISORT, and GHGSat instrument model enables us to simulate GHGSat measurements synthetically, serving as a benchmark. Additionally, we employ the same inverse model (Eq. 5) used in current GHGSat operational retrievals. Therefore, the OSSE results in this paper provide a truthful assessment of the simultaneous retrieval using the multi-angle viewing method across various aerosol and surface albedo conditions, demonstrating its direct applicability to GHGSat-like measurements. Discussions about applicability are added in Lines 244 - 250.

- Regarding the reviewer's comment that 'their simulations and retrievals were simplified by many fixed parameters and assumptions', our study conducted retrievals across a wide range of aerosol optical properties (SSA and g), surface albedo, and satellite zenith angle conditions to account for the diverse environmental conditions in observations. By fixing unrelated parameters and allowing the parameter of interest to vary during retrieval, we gain a deeper understanding of its impact. For example, to account for the considerable

variability in aerosol species in real-world scenarios, we presented example retrievals to quantify the mean bias and STD in retrieved $\Delta X_{CH_4}$, AOD, and $X_{alb}$ while varying aerosol SSA and g from 0 to 1. For demonstration purposes, we fixed the background surface albedo as 0.2, following Jervis et al. (2021). Results show that the simultaneous retrieval can significantly reduce the mean bias in $\Delta X_{CH_4}$ to within 0.3%. More importantly, the bias magnitude remains consistent across other surface albedo values, as indicated by Fig. 10 to Fig.11, indicating the robustness of our retrieval despite fixed parameters. Table 2 summarizes the performance of the retrieval method under different simulation configurations. While certain variables were fixed for demonstration purposes, the simultaneous retrieval exhibits consistently strong performance across diverse aerosol optical properties, surface albedo values, and satellite zenith angles. This indicates the retrieval's versatility across various simulation configurations.

- Regarding to the reviewer's suggestion to perform a sensitivity analysis for more complex simulation configurations, we appreciate this comment and have added additional results accordingly in Section 4.3. In retrieval, we assume near-surface aerosols with SSA = 0.95, g = 0.7, and sfc alb = 0.2. However, in the simulated GHGSat radiance, we introduce more complex (diverse) representations for both aerosol type (SSA and g), height distributions, and surface albedo. Details have been shown in the above discussions (Fig.S1-S3). The uncertainty in the mean bias and STD of $\Delta X_{CH_4}$ induced by assuming idealized aerosol types, height distributions, and surface albedo is less than 15.1% and 0.9%, respectively. These values suggest that the errors in simultaneous retrieval are relatively small, even in more complex 'real' measurements.

*(3) The numerical simulations in this paper are not able to represent the real retrievals because the real retrievals are more complex, so the conclusion of this paper is not necessarily correct. In particular, aerosol retrieval may introduce other uncertainties, but these are eliminated by other fixed parameters.*

- Regarding the reviewer's comment that 'numerical simulations in this paper are not able to represent the real retrievals', our study applies the same retrieval method (LFM in Eq.(5)) as current GHGSat instruments, which has been validated to successfully measure methane emissions across various real observations (Jervis et al., 2021; Maasakkers et al., 2022;

Jacob et al., 2022). Full GHGSat retrieval consists of two steps: a scene-wide retrieval to estimate the background average state vector $\widehat{X}$ and a per-cell retrieval to estimate the local methane plume enhancement. In this study, we focus on the per-cell retrieval assuming known background $\widehat{X}$ from the first step of our retrievals, which allows us to better detect the local methane enhancements. The simulated results in this paper provide a truthful assessment of applying the simultaneous retrieval technique to GHGSat-like point source imagers using the multi-angle viewing method. Retrieval experiments have been conducted across a wide range of aerosol optical properties, surface albedo, and satellite zenith angle conditions, demonstrating its direct applicability to real measurements. Discussions are added in Lines 246 - 250.

- Regarding the reviewer's comment that 'the aerosol retrieval may introduce other uncertainties', we have added the retrieval uncertainty analysis as suggested (see discussions above and section 4.3 in the manuscript). The retrieval uncertainties related to aerosol types, height distributions, and surface albedo are less than 15.1% for mean bias and 0.9% for standard deviations of $\Delta X_{CH_4}$. This indicates the robustness of the simultaneous retrieval method even under more different (complex) atmospheric conditions.

 *To summarise, this work could provide valuable information for methane measurements with the new instrument if the authors can improve the applicability of the retrieval algorithm.*

---

## Author Response (AR2)

**Author's response**

We truly appreciate the editor and reviewers for their helpful comments. Below are our responses (red) to their comments.

1) The technical issues identified by Reviewer #3;

- Figures throughout still use XCH4 throughout instead of ΔXCH4 as written in the text/captions
  XCH4 has been corrected as ΔXCH4 in all figure titles.
- There is still an instance of XAOD on L201
  XAOD has been corrected as AOD throughout the text.
- L281 and thereafter, ΔXalb is sometimes used in addition to Xalb - it is unclear to me if this is intentional and if so what the definition is
  ΔXalb has been corrected as Xalb throughout the text.

2) The depiction of the Inverse model (LFM) in Figure 4 appears to be misleading, as the term "inverse model" typically refers to a model or process utilized to address minimization (retrieval) issues. In this context, it is my understanding that LFM serves as a forward model used to simulate the radiances necessary for the minimization process.
Thank you for the thoughtful comment. To avoid confusion, we removed LFM from Figure 4.